# CONTINUITY-REGULARIZED FLOW MATCHING FOR OFFLINE REINFORCEMENT LEARNING

## ABSTRACT

Flow-matching policies have recently emerged as a powerful class of generative models for offline reinforcement learning (RL), capable of capturing complex, multi-modal action distributions from static datasets. However, standard training objectives are largely agnostic to the global properties of the generative path, permitting learned vector fields that are irregular and unstable, which can hinder performance. In this work, we introduce PDE-regularized Q-Learning (PQL), a novel algorithm that addresses this limitation by imposing a principled structure on the entire probability flow. PQL makes two synergistic contributions: first, a partial differential equation based regularizer derived from the continuity equation enforces global smoothness and stability on the flow. Second, to solve the complex optimization problem introduced by this regularizer, we propose a Beta-distributed timestep sampling strategy that focuses learning on the critical trajectory segments where the trade-off between imitation and smoothness is most acute. Through extensive experiments, we demonstrate that by structuring the generative journey and not just its destination, PQL achieves state-of-the-art performance on a wide range of challenging offline RL tasks.

## 1 INTRODUCTION

Recent advances in offline reinforcement learning (RL) have been driven by expressive generative policies capable of learning from static datasets with complex, multimodal action distributions (Wang et al., 2023; Zhu et al., 2023). Building on this progress, flow matching (FM) (Lipman et al., 2023) has emerged as a compelling alternative to diffusion models, delivering state-of-the-art performance on several offline RL benchmarks while offering a simpler, more efficient training paradigm (Park et al., 2025b; Zheng et al., 2023; Zhang et al., 2025). FM learns a time-dependent vector field defining an ordinary differential equation (ODE) that deterministically transports samples from a simple prior to a target action distribution, thus sidestepping the costly iterative sampling of diffusion-based approaches (Lipman et al., 2024).

Despite their effectiveness, current flow matching RL algorithms suffer from a fundamental, shared limitation. These methods learn the vector field $v_t(a)$ by regressing it onto a target field that defines a simple, direct path between a noise sample and a data sample (Lipman et al., 2023; Dao et al., 2023; Liu, 2022). The critical issue is that the standard loss objective evaluates this correspondence pointwise (Lipman et al., 2023). That is, it checks the accuracy of the vector field at randomly sampled points in time and space, but imposes no constraints on the relationship between vectors at different points. Consequently, the training objective is fundamentally path-agnostic (Liu et al., 2023; Tong et al., 2024): it enforces local correctness but leaves the global structure and smoothness of the generative path unconstrained, which constitutes a failure to learn or inaccurate learning from the underlying probability path (Huang et al., 2025), leading to a significant variation gap between the learned and ideal paths. In practice, such unconstrained flows manifest as training instability and the accumulation of numerical errors that degrade final policy performance.

We address this gap by introducing a principled regularizer derived from the governing physics of the generative process. The evolution of the density $p_t(a)$ under the learned velocity field $v_t(a)$ is precisely described by a continuity equation, a fundamental partial differential equation (PDE) from transport theory (Risken, 1989). Inspired by physics-informed machine learning (Cai et al., 2021; Raissi et al., 2019), we introduce a penalty that discourages violations of this continuity equation. By

enforcing this physical constraint, our regularizer ensures the model learns a smooth and consistent probability path, directly closing the variation gap and stabilizing the generative dynamics.

However, incorporating this PDE constraint introduces a new optimization challenge. The model must now learn a vector field that both matches the target data flow and satisfies the geometric constraints imposed by the regularizer. We argue that the standard practice of sampling time $t$ uniformly is not suited for this more complicated landscape. This is because the regularizer's influence is strongest not at the endpoints, but in the critical mid-range of the trajectory where an unconstrained model would learn its most complex and irregular dynamics Karras et al. (2022); Kingma et al. (2021). Uniform sampling under-allocates attention to these regions where the trade-off between imitation accuracy and physical consistency is most acute. To resolve this, we propose Beta-distributed time sampling, which focuses updates on this challenging regime to find a stable and effective solution for the PDE-constrained objective. Our contributions are threefold:

- We introduce a regularizer based on the continuity equation to address the unconstrained probability path in flow-matching policies, improving training stability by enforcing physical consistency on the generative dynamics.
- We design a Beta-distributed time sampling strategy that enables stable and efficient optimization of our PDE-constrained objective by focusing updates on the most challenging temporal regions of the flow.
- We demonstrate empirically that our method, which we call PQL, achieves state-of-the-art performance and significantly improves training stability on standard offline RL benchmarks.

## 2 RELATED WORK

**Diffusion Models for Offline RL.** Diffusion models (Hansen-Estruch et al., 2023; Ding et al., 2024; Chen et al., 2024a; Ding & Jin, 2024) have recently been adapted with great success to offline RL, offering a powerful tool for learning complex policies. Frameworks like Diffuser (Janner et al., 2022) treat decision-making as a trajectory denoising problem, generating entire plans that are then guided toward high-return outcomes. In robotics, diffusion-based planning has shown performance comparable to traditional methods while better capturing multi-modal behaviors (Chi et al., 2023; Chen et al., 2024b). Other approaches treat policies as conditional diffusion processes, iteratively refining actions from noise, which provides stable training and naturally handles multi-modal action distributions in high-dimensional spaces (Wang et al., 2023).

**Flow Matching for Offline RL.** Flow Matching (FM) has emerged as a highly efficient and deterministic alternative to diffusion for generative policy learning. FM models learn a vector field that defines an Ordinary Differential Equation (ODE) to transport samples from a simple prior to the target action distribution. This approach has demonstrated superior performance in continuous control tasks by leveraging the stability and efficiency of ODE-based generation(Lipman et al., 2023; 2024). In offline RL, FM has been applied to learn latent action spaces that ensure conservatism (Akimov et al., 2022) and to perform value-based guidance by weighting the flow objective with energy functions (Zhang et al., 2025). Flow-based methods are particularly beneficial for offline RL as their continuous and invertible nature helps avoid out-of-distribution actions while effectively utilizing the underlying data manifold (Park et al., 2025b).

## 3 PRELIMINARIES

### 3.1 GENERATIVE FLOW MATCHING

Flow matching (Liu et al., 2023) is a technique for training continuous-time generative models. The goal is to learn a parameterized vector field $u_\theta(x, t)$ that defines a probability flow $\{p_t\}_{t \in [0,1]}$ capable of transforming a simple prior distribution $p_0$ (e.g., a standard Gaussian $\mathcal{N}(0, I)$) into a complex data distribution $p_1$. The evolution of a sample $x$ over this flow is described by the ordinary differential equation (ODE):

$$\frac{dx_t}{dt} = u(x_t, t), \quad \text{with} \quad x_0 \sim p_0. \tag{1}$$

Generating a sample $x_1 \sim p_1$ is achieved by integrating this ODE from $t = 0$ to $t = 1$. The challenge lies in defining a suitable target vector field $u(x, t)$ for training.

Conditional Flow Matching (CFM) provides an elegant solution by defining a simple, fixed path between any pair of samples $x_0 \sim p_0$ and $x_1 \sim p_1$. A common choice is the linear interpolation path $x_t = (1 - t)x_0 + tx_1$. The target vector field along this path is simply its time derivative, which is constant: $v \triangleq \frac{dx_t}{dt} = x_1 - x_0$. The model $u_\theta(x_t, t)$ is then trained to regress onto this target field $v$ for all points $(x_t, t)$ along the path. This yields the flow matching loss:

$$\mathcal{L}_{\text{FM}}(\theta) = \mathbb{E}_{x_0 \sim p_0, x_1 \sim p_1, t \sim \mathcal{U}(0,1)} \big[ \| u_\theta((1 - t)x_0 + tx_1, t) - (x_1 - x_0) \|^2 \big]. \tag{2}$$

This objective allows for stable and efficient training of generative models without requiring simulations or adversarial training.

## 3.2 Offline Reinforcement Learning with Flow Matching

In offline reinforcement learning, the goal is to learn a high-performing policy from a static dataset $\mathcal{D} = \{(s, a, r, s')\}$ collected by one or more unknown behavior policies. We can frame policy learning as a generative modeling problem: we want to learn a conditional generator $\pi(a|s)$ that produces high-value actions.

Flow matching is a natural fit for this task. We learn a conditional vector field $u_\theta(s, a, t)$ that transforms a noise vector $a_0 \sim \mathcal{N}(0, I)$ into a desirable action $a_1$. To begin, we can train the model to simply imitate the actions in the dataset. This corresponds to a conditional flow matching objective where the target actions $a_1$ are drawn from the dataset $\mathcal{D}$:

$$\mathcal{L}_{\text{imitate}}(\theta) = \mathbb{E}_{(s,a_1) \sim \mathcal{D}, a_0 \sim \mathcal{N}(0,I), t \sim \mathcal{U}(0,1)} \big[ \| u_\theta(s, a_t, t) - (a_1 - a_0) \|^2 \big], \tag{3}$$

where $a_t = (1 - t)a_0 + ta_1$. This objective trains the policy to reproduce the action distribution $p_\mathcal{D}(a|s)$ found in the data. However, since the offline dataset is often suboptimal, pure imitation is insufficient. Modern approaches therefore augment this objective with value-based or reward-weighted terms to bias the learned policy toward actions that are expected to yield higher returns than those in the dataset. This framing provides the foundation for our proposed algorithm.

# 4 Methodology

Our proposed algorithm, PDE-regularized Q learning (PQL), enhances offline policy learning with flow matching by introducing two key innovations: (1) a PDE-guided regularizer to ensure stable and well-behaved policy flows, and (2) a novel beta-distribution-based timestep sampling strategy to focus the learning on critical parts of the generative trajectory.

## 4.1 PDE-Guided Regularized Policy Flows

As established in our preliminaries, we can train a conditional flow model to imitate the actions in an offline dataset using the objective $\mathcal{L}_{\text{imitate}}$ (eq. (3)). Building upon recent advancements in flow-based reinforcement learning (Park et al., 2025b), we move beyond simple imitation to improve upon the behavior policy. The base actor objective combines the imitation loss with a policy improvement term that encourages the flow to generate actions with high estimated Q-values via the following objective:

$$\mathcal{L}_{\text{imitate}}(\theta) + \mathbb{E}_{s \sim \mathcal{D}, a_0 \sim \mathcal{N}(0,I)} \big[ Q_\phi(s, a_1') \big], \tag{4}$$

where $a_1'$ is the action produced by integrating the learned vector field $u_\theta$ from $a_0$ over $t \in [0, 1]$. While this objective can learn an improved policy, it provides no explicit control over the underlying geometry of the probability flow. This can lead to instabilities or irregular dynamics, where the learned vector field is unnecessarily complex or sensitive to perturbations.

To address the potential for instability, we introduce our first contribution: a regularizer designed to enforce desirable geometric properties on the vector field $u_\theta$. Our approach is motivated by the continuity equation, a fundamental PDE that governs well-behaved physical flows.

For a fixed state $s$, the learned vector field $u_\theta(s, a, t)$ and its induced action distribution $\rho_t(a \mid s)$ must satisfy the continuity equation:

$$\partial_t \rho_t(a \mid s) + \nabla_a \cdot \big( \rho_t(a \mid s) \, u_\theta(s, a, t) \big) = 0. \tag{5}$$

This PDE ensures that the probability mass is conserved along the flow trajectories. However, directly enforcing Equation (5) by minimizing its squared residual—an objective of the form $\min \int \mathbb{E}_{a \sim \rho_t}[\mathcal{R}(s, a, t)^2]dt$ is intractable. This is because computing the residual term,

$$\mathcal{R}(s, a, t) \triangleq \partial_t \rho_t(a \mid s) + \nabla_a \cdot \big(\rho_t(a \mid s)\, u_\theta(s, a, t)\big), \tag{6}$$

**From Direct Enforcement to Sufficient Conditions.** Instead of tackling the intractable PDE directly, we propose to enforce simple sufficient conditions on the vector field $u_\theta$ that guarantee the existence, uniqueness, and stability of the solution to Equation (5). Standard results from transport theory (see Theorem 1) show that if $u_\theta$ is Lipschitz continuous in $a$ and has bounded divergence, the resulting density path $\rho_t(\cdot|s)$ is well-defined and stable.

To this end, we introduce a Jacobian-based regularizer that directly controls these properties:

$$\mathcal{L}_{\text{PDE}}(\theta) = \lambda\, \mathbb{E}_{s,a,t}\big[\|\nabla_a u_\theta(s, a, t)\|_F^2\big]. \tag{7}$$

Penalizing the Frobenius norm of the Jacobian is highly effective. It simultaneously bounds the Lipschitz constant of $u_\theta$ and its divergence. Unlike a simpler divergence-only penalty, our regularizer constrains all local deformations of the flow—including shear and rotation in addition to expansion and compression—which we find is crucial for mitigating instabilities and preventing the collapse of the action distribution during training.

**Theoretical Guarantees.** Our Jacobian-based regularizer is motivated by its ability to formally stabilize the learned probability flow. The following theorem provides a direct bound on the Wasserstein distance between the learned density path $\rho_t^\theta$ and a target path $\rho_t^\star$, explicitly in terms of the Jacobian norm that our method controls.

**Theorem 1** (Path Stability via Jacobian Control)**.** *Let $u_\star$ and $u_\theta$ be two vector fields. If the Jacobian of the learned field is bounded such that $\|\nabla_a u_\theta(s, a, t)\|_F \leq J$ for all $(s, a, t)$, and similarly for $u_\star$ with bound $J_\star$, then for any $t \in [0, 1]$, the corresponding density paths satisfy:*

$$W_2\big(\rho_t^\theta(\cdot \mid s), \rho_t^\star(\cdot \mid s)\big) \leq \exp\big((J + \sqrt{d}J)t\big) \int_0^t \Big(\mathbb{E}_{a \sim \rho_\tau^\star(\cdot|s)}[\|u_\theta(s, a, \tau) - u_\star(s, a, \tau)\|^2]\Big)^{1/2} d\tau, \tag{8}$$

*where $d$ is the action dimension.*

A brief proof is provided in Appendix C. The key insight is that bounding the Jacobian's Frobenius norm by $J$ provides an upper bound on both the Lipschitz constant ($L \leq J$) and the magnitude of the divergence ($|\nabla_a \cdot u_\theta| \leq \sqrt{d}J$). This theorem shows that by minimizing our regularizer $\mathcal{L}_{\text{PDE}}$ (Equation (7)), we directly tighten the stability bound on the policy flow.

Theorem 1 is a conditional stability statement: it only assumes that the Frobenius norm of the Jacobian of $u_\theta$ is uniformly bounded,

$$\sup_{s,a,t} \big\|\nabla_a u_\theta(s, a, t)\big\|_F \leq J < \infty.$$

No additional global Lipschitz assumption is required. For each $(s, a, t)$, the Lipschitz constant of $u_\theta$ with respect to $a$ is given by the operator norm of the Jacobian, which is always bounded by the Frobenius norm,

$$\big\|\nabla_a u_\theta(s, a, t)\big\|_{\text{op}} \leq \big\|\nabla_a u_\theta(s, a, t)\big\|_F.$$

Therefore, a bound on the Frobenius norm of the Jacobian provides a valid uniform upper bound on the Lipschitz constant that appears in the stability analysis. Our regularizer $\mathcal{L}_{\text{PDE}}$ is designed to directly control this bound during training.

The exponential dependence on $J$ in the bound of Theorem 1 is not specific to our method; it arises from standard applications of Grönwall's inequality in the stability analysis of ordinary differential equations. In practice, our flow time is restricted to $t \in [0, 1]$, so the factor $\exp\big((J + \sqrt{d}J)t\big)$ remains well behaved as long as $J$ is kept moderate. Without any regularization, the effective Jacobian norm can grow large and the bound becomes vacuous, which matches the unstable paths we observe empirically. By contrast, our Jacobian regularizer actively drives $J$ down during training, which tightens the bound and yields more stable generative paths.

To apply this regularization, the regularizer in Equation (7) is estimated efficiently without materializing the full Jacobian matrix by using Hutchinson's trace estimator. This method relies on Jacobian-vector products (JVPs), where the Jacobian is multiplied by a random probe vector $z$. We define $z$ as a vector sampled from a standard multivariate normal distribution, i.e., $z \sim \mathcal{N}(0, I)$, where $I$ is the identity matrix of the same dimension as the action space.

The JVP is the directional derivative of the vector field $u_\theta$ in the direction of $z$:

$$\mathrm{JVP}(u_\theta; z) \triangleq \nabla_a u_\theta(s, a, t) \cdot z$$

Modern deep learning frameworks compute this efficiently using forward-mode automatic differentiation. The squared Frobenius norm of the Jacobian is then recovered through the stochastic identity:

$$\|\nabla_a u_\theta(s, a, t)\|_F^2 = \mathbb{E}_{z \sim \mathcal{N}(0, I)} \|\mathrm{JVP}(u_\theta; z)\|^2. \tag{9}$$

In practice, a low-variance estimate is obtained using just one or two random probe vectors per sample, making the method scalable to high-dimensional action spaces.

**Regularized Actor Objective.** We integrate our PDE regularizer into the conditional flow-matching objective. Let $a_1 \sim p_\mathcal{D}(\cdot \mid s)$ be an action from the dataset, $a_0 \sim p_0$ be a sample from a base distribution, and $a_t = (1 - t)a_0 + ta_1$ be the linear interpolation with velocity $v = a_1 - a_0$. The actor is trained to match this velocity field while remaining regularized.

Crucially, to align the generative policy with the RL objective of maximizing returns, we use a learned critic $Q_\phi(s, a)$. The final regularized actor objective is:

$$\mathcal{L}_{\mathrm{actor}}(\theta) = \mathbb{E}_{s, a_0, a_1, t}\left[\|u_\theta(s, a_t, t) - (a_1 - a_0)\|^2\right] + \mathcal{L}_{\mathrm{PDE}}(\theta) + \mathbb{E}_{s, a_0}\left[Q_\phi(s, a_1')\right] + \mathcal{L}_{distill}, \tag{10}$$

where $a_1'$ is the action generated by integrating the learned field $u_\theta$ from $a_0$ at $t = 0$ to $t = 1$, the $\mathcal{L}_{distill}$ is the distillation term from (Park et al., 2025b) that encourages the one-step vector field prediction to align with the final generated action.

By minimizing our objective, we directly tighten the stability bound on the policy flow. However, adding this constraint to the learning objective without changing the training process can be suboptimal (We will demonstrate this in Section 5.4). By forcing the model to find a smoother solution, the regularizer makes the learning problem more challenging. A naive uniform sampling strategy may not provide a strong enough signal for the model to find a high-performing policy within this newly constrained landscape. This motivates our second contribution: an adaptive sampling strategy designed to provide the focused learning signal necessary to unlock the full potential of the regularized flow.

## 4.2 Adaptive Timestep Sampling via Beta Distribution

As we just motivated, the introduction of the $\mathcal{L}_{\mathrm{PDE}}$ regularizer fundamentally alters the training objective. The model is no longer tasked with simple imitation but must find a vector field $u_\theta$ that is both accurate to the target flow and geometrically simple, as measured by its Jacobian norm. This dual objective presents a non-uniform challenge across the trajectory $t \in [0, 1]$. The regularizer's smoothing effect is most impactful in regions where an unconstrained model would otherwise learn a complex field with high curvature or divergence. We posit that these critical regions typically occur mid-trajectory, as the flow dynamics are constrained by simpler boundary conditions (pure noise and clean data) at the endpoints, leaving the most complex geometric transformations for the intermediary steps.

A standard uniform sampling strategy, which allocates an equal training budget to all timesteps, is poorly matched to this non-uniform challenge. To address this, we build from the key insight that the standard $Unif(0, 1)$ distribution is simply a special case of the Beta distribution: $Unif(0, 1) \equiv Beta(1, 1)$. This provides a principled and simple family of distributions to explore.

**Proposition 1** (Optimal Sampling for Variance Reduction)**.** *Let* $\mathcal{L}(\theta) = \int_0^1 \ell(t)dt$ *be the total regularized loss. To minimize the variance of the stochastic gradient estimator for* $\nabla_\theta \mathcal{L}$*, the optimal sampling distribution* $\pi^*(t)$ *is proportional to the norm of the instantaneous gradient* $\pi^*(t) \propto \|\nabla_\theta \ell(t)\|$*.*

Based on this principle, we introduce adaptive sampling specifically to the imitation component of our objective. We focus sampling on the imitation loss as it is the primary driver of the flow's path,

while the $\mathcal{L}_{\mathrm{PDE}}$ term acts as a global constraint on the vector field's geometry, which we found benefits from uniform sampling across the entire trajectory.

Therefore, we define an adaptively sampled imitation loss, $\mathcal{L}$, which replaces the uniform sampling in the first part of the $\mathcal{L}_{\mathrm{actor}}$ term:

$$\mathcal{L}(\theta) = \mathbb{E}_{s,a_0,a_1,\,t\sim\mathrm{Beta}(\alpha,\alpha)} \left[ w_t^\pi \cdot \|u_\theta(s,a_t,t) - (a_1 - a_0)\|^2 \right], \tag{11}$$

where $t \sim \mathrm{Beta}(\alpha,\alpha)$ with $\alpha > 1$, and $w_t^\pi = \frac{t}{1-t}\pi(t)$ with Beta distribution density $\pi(t)$. This strategy focuses the model's capacity on the segments of the flow where the trade-off between imitation accuracy and PDE-enforced smoothness is most critical, leading to a more stable and efficient learning process.

## 5 EXPERIMENT

In this section, we empirically evaluate the performance of PQL, comparing it to previous offline RL approaches on a variety of challenging tasks. We also provide extensive analyses and ablations on the effectiveness of different components and PQL's design choices. Extra experiments can be found in Appendix E.

### 5.1 EXPERIMENTAL SETUP

We conduct a comprehensive evaluation of our proposed algorithm on D4RL(Fu et al., 2020) and OGBench(Park et al., 2025a) benchmark, following the standard experimental protocols established in prior work (Park et al., 2025b). Our evaluation spans a diverse suite of tasks designed to test capabilities in locomotion, navigation, and complex manipulation. To ensure a fair comparison across environments with different reward scales, we use the standard D4RL normalized score as our primary performance metric (Fu et al., 2020). All reported results are averaged with standard deviations over eight random seeds to ensure statistical robustness.

**Baselines.** We benchmark our algorithm against a comprehensive set of state-of-the-art methods spanning three major categories: traditional, diffusion-based, and flow-based offline RL. To ensure a fair and robust comparison, we select specific baselines for different task suites based on the availability of established implementations in the literature (Hu et al., 2025). For the **D4RL Gym-MuJoCo** benchmarks, we compare against **traditional methods** including BC, IQL (Kostrikov et al., 2022), and CQL (Kumar et al., 2020); **diffusion-based methods** such as IDQL (Hansen-Estruch et al., 2023), SRPO (Chen et al., 2024a), and CAC (Ding & Jin, 2024); and the **flow-based methods** FQL (Park et al., 2025b), Flow (Zheng et al., 2023), and CNF (Akimov et al., 2022). For the more complex **Adroit** and **OGBench** manipulation tasks, our comparison suite includes a different set of established baselines, with the flow-based methods comprising FAWAC, FBRAC (Park et al., 2025b).

### 5.2 OFFLINE EVALUATION

Our proposed algorithm, PQL, establishes a new state-of-the-art on the D4RL Gym-MuJoCo benchmarks, as detailed in Table 1. PQL's strength is particularly pronounced in datasets containing high-quality trajectories, achieving the top or tied-for-top score in five of the six Medium-Expert and Medium-Replay settings. This dominance suggests that the synergy between our PDE regularizer and adaptive sampling is highly effective at structuring the policy around expert data, enabling the model to stably and efficiently learn high-reward behaviors. When compared to leading algorithms, PQL consistently surpasses prior flow-based methods, while also demonstrating a clear advantage over strong diffusion-based models like SRPO and CNF in the most challenging 'Medium-Expert' tasks. Furthermore, PQL remains highly competitive in the noisier 'Medium' datasets, securing the highest score on Walker2d and demonstrating robust performance on Hopper. These results validate our core hypothesis that structuring the entire generative path, not just its endpoint, leads to more performant offline RL agents.

On the challenging Adroit manipulation tasks, which feature a high-dimensional state-action space and sparse rewards, PQL demonstrates a significant performance advantage over prior methods, as shown in Table 2. Our method achieves state-of-the-art results in 11 out of the 12 task settings,

Table 1: Offline RL algorithms comparison on D4RL Gym-MuJoCo environments, grouped by task. Red indicates the best results while blue indicates the second best.

| | HalfCheetah | | | Hopper | | | Walker2d | | |
|---|---|---|---|---|---|---|---|---|---|
| Algorithm | Medium-Expert | Medium | Medium-Replay | Medium-Expert | Medium | Medium-Replay | Medium-Expert | Medium | Medium-Replay |
| BC | 55.2 | 42.6 | 36.6 | 52.5 | 52.9 | 18.1 | 107.5 | 75.3 | 26.0 |
| IQL | 93±3 | 50±0.2 | 42±4 | 86±30 | 65±4 | 90±13 | 112±0.5 | 81±3 | 75±9 |
| CQL | 62±3 | 45±2 | 47±2 | 99±5 | 58±3 | 49±2 | 110±5 | 79±4 | 27±2 |
| IDQL | 94±3 | 50±1 | 45±0.8 | 105±3 | 63±2 | 82±10 | 112±0.7 | 71±10 | 82±3 |
| SRPO | 92±3 | 60±0.8 | 51±3 | 100±14 | 95±2 | 101±1 | 114±2 | 84±4 | 84±7 |
| CAC | 59±4 | 69±0.7 | 59±4 | 100±4 | 81±11 | 99±0.5 | 110±0.7 | 81±10 | 79±4 |
| FQL | 86.1±6 | 60±3 | 53±6 | 21±3 | 25±0.8 | 28±3 | 13±10 | 9±2 | 7±3 |
| Flow | 97±1 | 49±2 | 42±0.8 | 105±3 | 84±2 | 89±6 | 94±1 | 77±2 | 78±0.9 |
| CNF | 96±0.3 | 51±0.5 | 46±0.2 | 109±5 | 69.3±1 | 89±10 | 112±0.5 | 84±3 | 82±2 |
| PQL(Ours) | 90±1 | 54±2 | 51±0.3 | 112±3 | 84±1 | 92±3 | 114±0.8 | 86±3 | 84±1 |

showcasing its effectiveness in complex, human-centric domains. PQL's performance is particularly dominant on the cloned and expert datasets, where it consistently achieves the highest scores across all four environments. This suggests that the structural priors from our PDE regularizer and the efficiency of Beta sampling are highly effective at extracting and stabilizing the complex skills present in high-quality demonstration data.

Table 2: Algorithm comparison across Adroit environments, grouped by task. Red indicates the best results while blue indicates the second best.

| | Pen | | | Door | | | Hammer | | | Relocate | | |
|---|---|---|---|---|---|---|---|---|---|---|---|---|
| Algorithm | Human | Cloned | Expert | Human | Cloned | Expert | Human | Cloned | Expert | Human | Cloned | Expert |
| BC | 34 | 57 | 85 | 0.5 | -0.1 | 35 | 2 | 0.8 | 126 | 0.0 | -0.1 | 101 |
| IQL | 82±18 | 77±18 | 134±16 | 3±2 | 0.8±1 | 105±3 | 3±2 | 1±0.5 | 130±0.5 | 0.1±0.1 | 0.2±0.4 | 107±3 |
| CQL | 38±2 | 39±5 | 107±9 | 10±2 | 0.4±0.2 | 101±2 | 4±1 | 2±1 | 87±5 | 0.2±0.1 | -0.1±0.1 | 95±5 |
| IDQL | 76±10 | 64±7 | 140±6 | 6±2 | 0±0 | 105±1 | 2±1 | 2±1 | 125±4 | 0±0 | -0±0 | 107±1 |
| SRPO | 69±7 | 61±7 | 134±4 | 3±3 | 0±0 | 105±0.5 | 1±1 | 2±1 | 127±0 | 0±0 | -0±0 | 106±2 |
| CAC | 64±8 | 56±10 | 103±9 | 5±2 | 1±0 | 98±3 | 2±0 | 1±1 | 92±11 | 0±0 | -0±0 | 93±6 |
| FAWAC | 67±5 | 62±10 | 118±6 | 2±1 | 0±1 | 103±1 | 2±1 | 1±0 | 118±3 | 0±0 | -0±0 | 105±3 |
| FBRAC | 77±7 | 67±9 | 119±7 | 4±2 | 0±0 | 104±2 | 2±1 | 2±1 | 119±9 | 0±0 | 1±1 | 105±2 |
| IFQL | 71±12 | 80±11 | 139±5 | 7±2 | 2±2 | 104±2 | 3±1 | 2±1 | 117±9 | 0±0 | -0±0 | 104±3 |
| FQL | 53±16 | 74±11 | 142±6 | 0±0 | 2±1 | 104±1 | 1±1 | 11±9 | 125±3 | 0±0 | -0±0 | 107±1 |
| PQL(Ours) | 82±3 | 87±6 | 148±7 | 4±1 | 3±0.8 | 107±2 | 4±1 | 14±6 | 134±1 | 0.8±0.1 | 0.5±0.1 | 110±1 |

Furthermore, PQL delivers highly competitive performance on the difficult human datasets, which are known for their suboptimality and noise. It secures the second-best score on the 'pen-human' task and demonstrates strong results on 'hammer-human', outperforming the majority of prior diffusion and flow-based methods. This robust performance across all data qualities highlights the general applicability of our approach. By enforcing a smooth and stable generative process, PQL is able to learn effective manipulation policies even from challenging and imperfect data, setting a new standard for flow-based offline RL in high-dimensional control.

Our analysis extends to the challenging OGBench suite, which includes complex navigation and manipulation tasks designed to test for generalization. As demonstrated in Table 3, PQL consistently achieves state-of-the-art performance, securing the top score in 8 out of the 10 evaluation settings. This strong result across a diverse set of environments underscores the versatility and robustness of our proposed method. In the difficult long-horizon navigation tasks of Antmaze and Humanoid, PQL significantly outperforms prior methods, suggesting that the smooth and stable flows learned via our PDE regularizer are critical for avoiding local optima and discovering effective long-range policies. Furthermore, PQL excels in the dexterous manipulation tasks within Cube, Scene, and Puzzle, where it surpasses all baselines by a large margin. This indicates that our adaptive Beta sampling strategy is highly effective at capturing the fine-grained, multi-modal behaviors required for successful manipulation. The consistent, top-tier performance across these varied domains validates our approach of structuring the entire generative path, proving it to be a powerful principle for building generalist offline RL agents.

Table 3: Algorithm comparison on selected OGBench tasks, grouped by environment. Red indicates the best results while blue indicates the second best.

| | Antmaze | | Humanoid | | Antsoccer | Cube | | Scene | Puzzle | |
|---|---|---|---|---|---|---|---|---|---|---|
| Algorithm | Large-nav | Giant-nav | Medium-nav | Large-nav | Arena-nav | Single-play | Double-play | Play | 3x3-Play | 4x4-Play |
| BC | 0±0 | 0±0 | 1±0 | 0±0 | 1±0 | 2±1 | 0±0 | 1±1 | 1±1 | 0±0 |
| IQL | 48±20 | 0±0 | 32±7 | 0±0 | 3±2 | 80±8 | 1±0 | 12±8 | 2±1 | 5±2 |
| CQL | 23±12 | 0±0 | 20±4 | 0±0 | 1±1 | 82±7 | 2±1 | 8±2 | 1±1 | 4±2 |
| IDQL | 0±0 | 0±0 | 1±2 | 0±0 | 5±2 | 96±1 | 16±10 | 30±14 | 3±3 | 26±6 |
| SRPO | 0±0 | 0±0 | 0±0 | 0±0 | 20±7 | 96±2 | 0±0 | 2±2 | 0±0 | 7±8 |
| CAC | 42±57 | 0±0 | 38±19 | 1±0 | 0±0 | 88±12 | 2±2 | 32±18 | 1±0 | 1±0 |
| FAWAC | 1±1 | 0±0 | 6±2 | 0±0 | 12±3 | 81±10 | 2±1 | 18±6 | 1±2 | 0±0 |
| FBRAC | 70±20 | 0±0 | 25±6 | 0±0 | 24±4 | 83±8 | 22±12 | 46±20 | 2±2 | 5±3 |
| IFQL | 24±17 | 1±0 | 69±11 | 6±12 | 16±9 | 79±5 | 9±15 | 0±0 | 0±0 | 21±11 |
| FQL | 80±32 | 4±4 | 19±13 | 7±2 | 39±36 | 98±3 | 36±6 | 76±9 | 16±5 | 11±3 |
| PQL(Ours) | 84±7 | 7±1 | 30±10 | 9±1 | 41±15 | 100±1 | 41±4 | 80±3 | 19±3 | 16±1 |

## 5.3 DESIGN CHOICE

**Jacobian Regularization vs. Divergence Penalty** A core contribution of our work is the PDE-guided regularizer, which penalizes the squared Frobenius norm of the Jacobian ($\|\nabla_a u_\theta\|_F^2$). A simpler alternative would be to penalize only the divergence of the vector field ($\nabla_a \cdot u_\theta$) as suggested in prior work (Huang et al., 2025). Our choice to penalize the full Jacobian is deliberate, as it provides a more comprehensive and robust form of regularization critical for policy stability by constraining not just the expansion and compression of the flow, but also its rotational and shear dynamics.

To empirically validate this choice, we conduct an experiment comparing our full PQL algorithm against a variant using a divergence-only penalty, denoted "PQL with Div". The learning curves across the nine D4RL Gym-MuJoCo environments are presented in Figure 1. The results clearly demonstrate the superiority of the full Jacobian penalty. While the divergence-only regularizer offers some benefit, our full PQL model consistently achieves higher final performance and often exhibits faster convergence across all three environments, particularly in the more challenging 'Medium-Expert' and 'Medium-Replay' datasets. For example, in 'Hopper-MR' and 'Walker2d-ME', our method establishes a clear and stable performance gap over the divergence-only variant throughout training.

The most significant advantage is observed in the stability and asymptotic performance of our method. The learning curves for PQL are generally smoother with narrower error bands, indicating lower variance and more reliable convergence. In contrast, "PQL with Div" often plateaus at a lower score, suggesting that controlling only the divergence is insufficient to prevent the policy from converging to a suboptimal solution. This confirms our hypothesis that constraining the complete local geometry of the flow is crucial for learning high-performing and robust policies.

## 5.4 ABLATION STUDY

To dissect the individual contributions of our two primary components—the PDE-guided regularizer and the adaptive Beta sampling—we conduct a targeted ablation study. We evaluate three distinct variants: PQL (Ours), the full proposed algorithm with both components; PQL-Beta, a variant using the PDE regularizer but with standard uniform timestep sampling; and PQL-PDE, a variant using our adaptive Beta sampling but without the PDE regularizer.

Table 4: Ablation study of PQL components on D4RL Gym-MuJoCo environments. We compare our full method against variant without the PDE regularizer (PQL-PDE) and without Beta sampling (PQL-Beta).

| | HalfCheetah | | | Hopper | | | Walker2d | | |
|---|---|---|---|---|---|---|---|---|---|
| Algorithm | Medium-Expert | Medium | Medium-Replay | Medium-Expert | Medium | Medium-Replay | Medium-Expert | Medium | Medium-Replay |
| PQL-Beta (w/o Beta) | 76.2±5.5 | 39.5±2.8 | 48.1±1.5 | 98.7±4.1 | 75.3±2.5 | 85.0±4.2 | 105.6±2.1 | 79.8±4.0 | 78.5±2.9 |
| PQL-PDE (w/o PDE) | 84.3±4.9 | 45.1±3.5 | 45.2±2.4 | 102.4±3.8 | 79.5±1.9 | 88.6±3.9 | 109.8±1.9 | 82.0±3.5 | 80.1±2.1 |
| PQL | 90.2±1.4 | 54.1±2.2 | 51.3±0.3 | 111.8±2.9 | 84.2±1.1 | 92.1±3.4 | 114.4±0.8 | 86.2±3.1 | 84.4±1.2 |

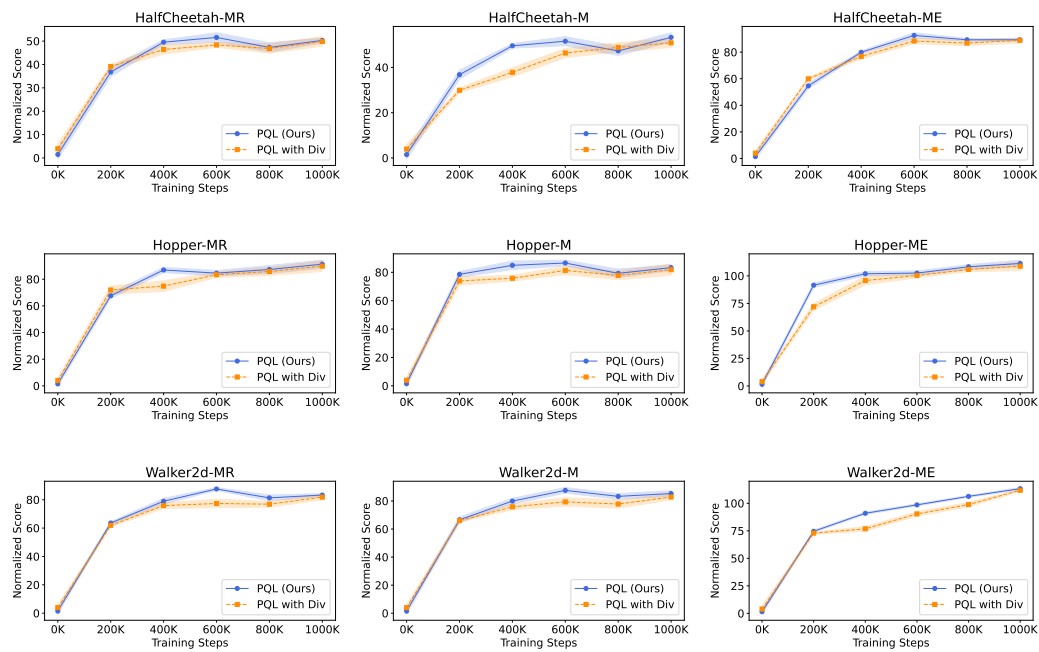

Figure 1: Performance comparison for different design choices of PDE on the D4RL Gym-MuJoCo environments. Blue line indicates the proposed PQL while orange line indicates the variants with divergence match.

Our ablation study, presented in Table 4 and Figure 2, provides critical insights into the interplay between our proposed components. The performance of the PQL-Beta variant, which applies the PDE regularizer with standard uniform time sampling, is particularly revealing. This model struggles across several environments, suggesting that adding the PDE constraint without a more efficient learning mechanism can be detrimental. While the regularizer successfully creates a smoother optimization landscape, the unfocused updates from uniform sampling appear insufficient for the policy to find a high-performing solution within this more challenging, constrained space. In contrast, the PQL-PDE variant, which uses adaptive Beta sampling without the regularizer, achieves strong results. This indicates that focusing the training on the most informative parts of the generative trajectory is a powerful mechanism for improving performance, even without explicit regularization of the vector field's geometry.

### 5.5 HYPER-PARAMETER

To analyze the sensitivity of PQL to the Beta distribution's shape parameter $\alpha$, we conduct a hyperparameter sweep on three representative D4RL Gym-MuJoCo environments. The results, shown in Figure 3, demonstrate a clear and consistent trend across all three tasks. Performance is lowest at $\alpha = 1.0$ (the uniform sampling baseline) and increases significantly for any $\alpha > 1$. The optimal performance is consistently achieved within the range of $\alpha \in [2.0, 5.0]$, with a slight decrease for larger values, likely due to over-focusing on the trajectory's midpoint. This study confirms that our method is robust to the precise choice of this parameter and validates our use of a default value of $\alpha = 3.0$ for majority of the experiments.

## 6 CONCLUSION

We introduced PQL, a novel flow-based algorithm that improves the stability and efficiency of generative policies in offline RL. We identified that standard flow-matching objectives are path-agnostic, which can lead to irregular vector fields and inefficient training. Our solution is a synergistic, two-part approach: a PDE-guided regularizer that enforces a smooth, physically coherent probability

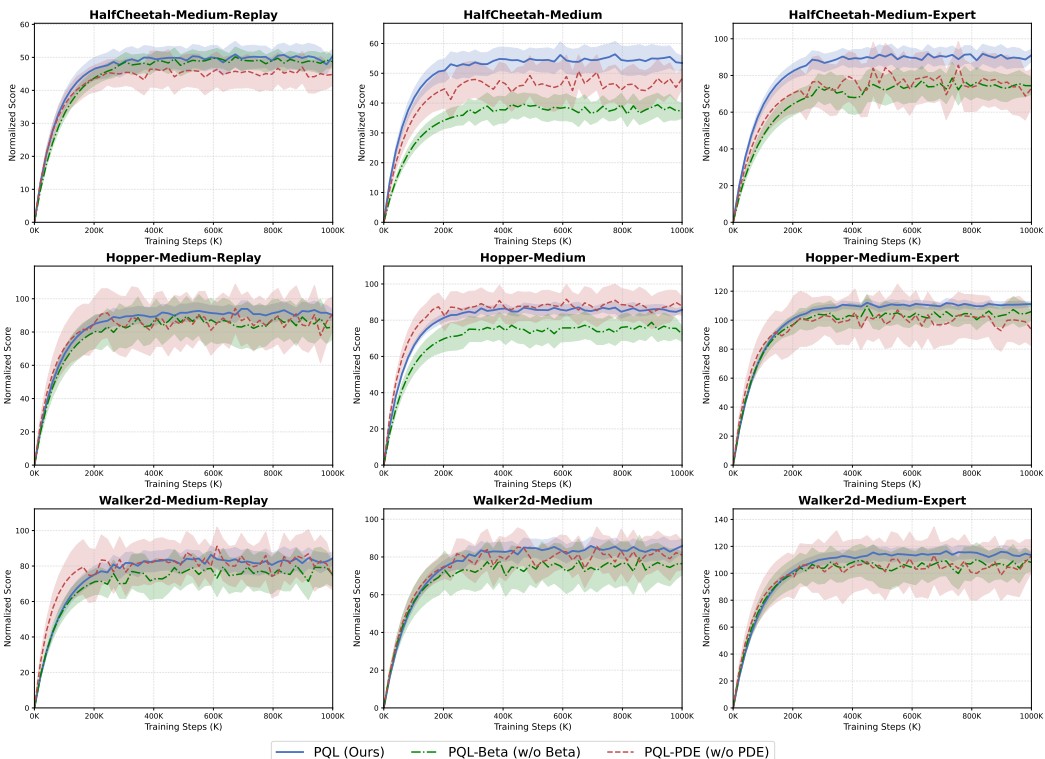

Figure 2: Training Curve on D4RL Gym-MuJoCo environments across 8 different seeds. Blue line indicates the proposed PQL while the red indicates the variant without PDE regularizer, the green line indicates the variant without Beta.

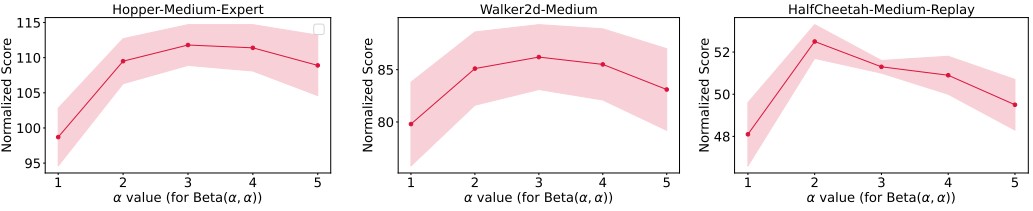

Figure 3: Hyper-parameter study

flow, and an adaptive Beta-distributed time sampling strategy that resolves the optimization challenges of this regularizer by focusing updates on the most informative parts of the generative trajectory. Our extensive experiments on the D4RL, Adroit, and OGBench benchmarks demonstrate that PQL achieves state-of-the-art performance, particularly in complex, high-dimensional control tasks. The results validate that our approach of structuring the entire generative path leads to more robust and sample-efficient policy learning. Future work could explore extending these principles to other generative model families or investigating more sophisticated, state-dependent time sampling curricula.

While PQL demonstrates strong performance, it has two primary limitations. First, our PDE-regularizer introduces a modest computational overhead, as it requires $K$ additional Jacobian-vector products (JVPs) per training step. Second, our method adds two key hyperparameters, the regularization weight $\lambda$ and the sampling parameter $\alpha$. While our extensive analysis shows these parameters are robust and consistent across a wide range of tasks, they still require initial tuning

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

## A USAGE OF LLM

We declare that the LLM is used for polishing writing only.

## B    CONNECTION TO THE FOKKER-PLANCK EQUATION

In this section, we detail the connection between the Ordinary Differential Equation (ODE) that defines our generative flow, the corresponding stochastic process, and the Fokker-Planck Equation (FPE) that governs the evolution of its probability density. This connection provides the theoretical grounding for our PDE-guided regularizer.

### B.1    FROM ODES TO SDES AND PROBABILITY FLOWS

Our flow-matching policy learns a deterministic process governed by an ODE:

$$da_t = u_\theta(s, a_t, t)dt, \tag{12}$$

where $u_\theta$ is the learned vector field. This describes the trajectory of a single sample. To understand the evolution of the entire probability density $p_t(a|s)$, it is useful to consider the corresponding stochastic differential equation (SDE). A deterministic ODE like ours can be seen as a special case of an SDE with zero diffusion:

$$da_t = u_\theta(s, a_t, t)dt + \sigma(t)dW_t, \quad \text{where} \quad \sigma(t) = 0. \tag{13}$$

Here, $W_t$ is a standard Wiener process (Brownian motion). This SDE formulation provides a powerful bridge to the language of probability flows.

### B.2    THE FOKKER-PLANCK EQUATION

For any process described by an SDE of the form $dx_t = f(x_t, t)dt + g(x_t, t)dW_t$, the evolution of its probability density $p(x, t)$ is governed by a partial differential equation known as the Fokker-Planck Equation (also called the forward Kolmogorov equation):

$$\frac{\partial p(x, t)}{\partial t} = -\nabla_x \cdot [f(x, t)p(x, t)] + \frac{1}{2}\nabla_x^2 \cdot [g(x, t)^2 p(x, t)]. \tag{14}$$

The first term on the right-hand side is the drift term, describing how the mean of the distribution is pushed by the vector field. The second is the diffusion term, describing how the distribution spreads out due to noise.

In our specific case, the process is deterministic. The drift is given by our learned vector field, $f(a_t, t) = u_\theta(s, a_t, t)$, and the diffusion is zero, $g(a_t, t) = 0$. When we substitute these into the general FPE, the entire diffusion term vanishes. This leaves us with:

$$\frac{\partial p_t(a|s)}{\partial t} = -\nabla_a \cdot [u_\theta(s, a, t)p_t(a|s)]. \tag{15}$$

Rearranging this equation yields the continuity equation, which we use to motivate our regularizer in the main paper:

$$\frac{\partial p_t(a|s)}{\partial t} + \nabla_a \cdot [u_\theta(s, a, t)p_t(a|s)] = 0. \tag{16}$$

This derivation formally shows that the continuity equation is the specific instance of the Fokker-Planck Equation that governs the probability flow of a deterministic, ODE-based generative model. Therefore, our PDE-guided regularizer is not an arbitrary choice but is directly derived from the fundamental physics of probability flows.

## C    PROOF OF THEOREM 1

We begin by stating the integral form of Grönwall's inequality, which is a foundational result for bounding functions that satisfy certain differential inequalities.

**Lemma 1** (Grönwall's Inequality). *Let $y(t)$ and $\alpha(t)$ be non-negative continuous functions for $t \geq 0$, and let $\beta \geq 0$ be a constant. If $y(t)$ satisfies:*

$$y(t) \leq \beta + \int_0^t \alpha(s)y(s)\,ds$$

*Then, for all $t \geq 0$, $y(t)$ is bounded by:*

$$y(t) \leq \beta \exp\left(\int_0^t \alpha(s)\,ds\right)$$

This lemma is essential for deriving bounds on quantities that evolve over time, such as the Wasserstein distance between two probability flows.

Next, we formally establish the connection between the Frobenius norm of the Jacobian, $\|\nabla_a u\|_F$, and the two key properties of the vector field: its Lipschitz constant $L$ and the magnitude of its divergence $|\nabla_a \cdot u|$.

**Lemma 2** (Jacobian Norm Bounds). *Let* $u : \mathbb{R}^d \to \mathbb{R}^d$ *be a differentiable vector field. If* $\|\nabla_a u(a)\|_F \leq J$ *for all $a$, then:*

    *1. $u$ is $J$-**Lipschitz continuous***.

    *2. The divergence is bounded:* $|\nabla_a \cdot u(a)| \leq \sqrt{d}J$.

*Proof.* **1. Lipschitz Constant:** By the Mean Value Theorem for vector-valued functions, for any two points $a_1, a_2 \in \mathbb{R}^d$:

$$\|u(a_1) - u(a_2)\| \leq \sup_{c \in [a_1, a_2]} \|\nabla_a u(c)\|_{op} \|a_1 - a_2\|$$

where $\|\cdot\|_{op}$ is the operator norm (largest singular value). We know the operator norm is bounded by the Frobenius norm: $\|\mathbf{A}\|_{op} \leq \|\mathbf{A}\|_F$. Since $\|\nabla_a u(a)\|_F \leq J$ for all $a$, we have:

$$\|u(a_1) - u(a_2)\| \leq J\|a_1 - a_2\|$$

Thus, $u$ is $J$-Lipschitz.

**2. Divergence Bound:** The divergence is the trace of the Jacobian: $\nabla_a \cdot u = \mathrm{Tr}(\nabla_a u)$. Let $\lambda_1, \ldots, \lambda_d$ be the eigenvalues of $\nabla_a u$. Then $\mathrm{Tr}(\nabla_a u) = \sum_{i=1}^d \lambda_i$. By the Cauchy-Schwarz inequality:

$$|\nabla_a \cdot u|^2 = \left| \sum_{i=1}^d \lambda_i \right|^2 \leq \left( \sum_{i=1}^d |\lambda_i|^2 \right) \left( \sum_{i=1}^d 1^2 \right) = d \sum_{i=1}^d |\lambda_i|^2$$

Schur's inequality states that $\sum_{i=1}^d |\lambda_i|^2 \leq \|\nabla_a u\|_F^2$. Therefore:

$$|\nabla_a \cdot u|^2 \leq d\|\nabla_a u\|_F^2 \leq dJ^2$$

Taking the square root gives $|\nabla_a \cdot u| \leq \sqrt{d}J$. This completes the proof of the lemma. $\square$

*Proof.* We now prove the main theorem. Let $\rho_t^\theta$ and $\rho_t^\star$ be the probability densities induced by the vector fields $u_\theta$ and $u_\star$, respectively, starting from the same initial distribution $\rho_0$. A standard result for flows (see, e.g., Theorem 5.34 in Santambrogio's "Optimal Transport for Applied Mathematicians") states that the 2-Wasserstein distance $y(t) = W_2(\rho_t^\theta, \rho_t^\star)$ satisfies the differential inequality:

$$\frac{d}{dt} y(t) \leq Ly(t) + \left( \mathbb{E}_{a \sim \rho_t^\star}[\|u_\theta(a, t) - u_\star(a, t)\|^2] \right)^{1/2}$$

where $L$ is an upper bound on the Lipschitz constants of the fields. A more refined bound, considering both expansion and contraction, is:

$$\frac{d}{dt} y(t) \leq (L + D)y(t) + E(t)$$

where $D$ bounds the divergence and $E(t) = \left( \mathbb{E}_{a \sim \rho_t^\star}[\|u_\theta(a, t) - u_\star(a, t)\|^2] \right)^{1/2}$.

From Lemma 2, we can set $L = J$ and $D = \sqrt{d}J$ (assuming $J$ is an upper bound for both fields). This gives the differential inequality:

$$\frac{d}{dt} y(t) \leq (J + \sqrt{d}J)y(t) + E(t)$$

Since $y(0) = W_2(\rho_0, \rho_0) = 0$, applying Grönwall's inequality (or solving the associated linear ODE) yields:

$$y(t) \leq \int_0^t E(\tau) \exp\left( \int_\tau^t (J + \sqrt{d}J)ds \right) d\tau$$

$$y(t) \leq \int_0^t E(\tau) \exp\left((J + \sqrt{d}J)(t - \tau)\right) d\tau$$

Since $t - \tau \leq t$ for $\tau \in [0, t]$, we can find a looser but simpler bound by taking the maximal value of the exponential term outside the integral:

$$W_2(\rho_t^\theta, \rho_t^\star) \leq \exp((J + \sqrt{d}J)t) \int_0^t \left(\mathbb{E}_{a \sim \rho_\tau^\star}[\|u_\theta(s, a, \tau) - u_\star(s, a, \tau)\|^2]\right)^{1/2} d\tau$$

This completes the proof of the theorem. $\qquad\square$

## D    PROOF OF PROPOSITION 1

*Proof.* The proof follows the standard procedure for finding an optimal importance sampling distribution. The goal is to minimize the variance of an unbiased estimator subject to the constraint that the sampling distribution is a valid probability density function.

We wish to estimate the total gradient of the loss, which is an integral over the time variable $t$:

$$G(\theta) = \nabla_\theta \mathcal{L}(\theta) = \nabla_\theta \int_0^1 \ell(t)dt = \int_0^1 \nabla_\theta \ell(t)dt$$

We can form a stochastic, single-sample Monte Carlo estimator for this integral by sampling a timestep $t$ from a proposal distribution $\pi(t)$ and weighting the result. The importance sampling estimator for the gradient $G(\theta)$ is:

$$\hat{G}(t) = \frac{\nabla_\theta \ell(t)}{\pi(t)}$$

This is an **unbiased estimator**, as its expectation is equal to the true gradient:

$$\mathbb{E}_{t \sim \pi(t)}[\hat{G}(t)] = \int_0^1 \frac{\nabla_\theta \ell(t)}{\pi(t)} \pi(t)dt = \int_0^1 \nabla_\theta \ell(t)dt = G(\theta)$$

Our goal is to choose the distribution $\pi(t)$ that minimizes the variance of this estimator. The variance of a vector estimator is the expected squared Euclidean distance from its mean. Minimizing this is equivalent to minimizing $\mathbb{E}[\|\hat{G}(t)\|^2]$, since the mean $G(\theta)$ is fixed with respect to $\pi(t)$. The second moment is:

$$\mathbb{E}_{t \sim \pi(t)}[\|\hat{G}(t)\|^2] = \int_0^1 \left\|\frac{\nabla_\theta \ell(t)}{\pi(t)}\right\|^2 \pi(t)dt = \int_0^1 \frac{\|\nabla_\theta \ell(t)\|^2}{\pi(t)} dt$$

So, our optimization problem is to find the function $\pi(t)$ that minimizes this integral.

We must minimize the variance expression subject to the constraint that $\pi(t)$ is a valid probability distribution, meaning $\int_0^1 \pi(t)dt = 1$. This is a constrained optimization problem that we can solve using a **Lagrange multiplier** $\lambda$. The Lagrangian $\mathcal{J}$ is:

$$\mathcal{J}(\pi, \lambda) = \int_0^1 \frac{\|\nabla_\theta \ell(t)\|^2}{\pi(t)} dt + \lambda \left(\int_0^1 \pi(t)dt - 1\right)$$

To find the optimal $\pi(t)$, we take the functional derivative of $\mathcal{J}$ with respect to $\pi(t)$ and set it to zero:

$$\frac{\delta \mathcal{J}}{\delta \pi(t)} = -\frac{\|\nabla_\theta \ell(t)\|^2}{\pi(t)^2} + \lambda = 0$$

Solving for $\pi(t)$, we find:

$$\pi(t)^2 = \frac{\|\nabla_\theta \ell(t)\|^2}{\lambda} \implies \pi(t) = \frac{\|\nabla_\theta \ell(t)\|}{\sqrt{\lambda}}$$

This shows that the optimal distribution $\pi(t)$ must be proportional to the norm of the instantaneous gradient $\|\nabla_\theta \ell(t)\|$.

We find the value of the constant $\sqrt{\lambda}$ by enforcing the constraint $\int_0^1 \pi(t)dt = 1$:

$$\int_0^1 \frac{\|\nabla_\theta \ell(t)\|}{\sqrt{\lambda}} dt = 1$$

$$\frac{1}{\sqrt{\lambda}} \int_0^1 \|\nabla_\theta \ell(t)\| dt = 1 \implies \sqrt{\lambda} = \int_0^1 \|\nabla_\theta \ell(\tau)\| d\tau$$

Let $Z = \int_0^1 \|\nabla_\theta \ell(\tau)\| d\tau$ be the normalization constant. Substituting this back, we get the full form of the optimal distribution:

$$\pi^*(t) = \frac{\|\nabla_\theta \ell(t)\|}{Z} = \frac{\|\nabla_\theta \ell(t)\|}{\int_0^1 \|\nabla_\theta \ell(\tau)\| d\tau}$$

This shows that the sampling distribution $\pi^*(t)$ that minimizes the variance of the stochastic gradient estimate is the one that is proportional to the magnitude of the gradient being estimated at each point $t$. $\qquad\square$

## E    EXTRA EXPERIMENTS

### E.1    DESIGN CHOICE

We also provide the extra experiments regarding the design choice in Adroit and OGBench. Figure 4 presents the extra ablation study comparing our proposed method, **PQL (Ours)**, against a key variant, **PQL with Divergence**. This variant replaces our continuity-based regularizer with a more standard divergence-based penalty, allowing us to isolate the benefits of enforcing a physically consistent generative path. The comparison is conducted across two distinct and challenging benchmark suites: Adroit and OGBench.

Across the Adroit manipulation tasks (Figure 4a), PQL demonstrates **consistently superior performance**. In all 12 environments, our method achieves a higher mean normalized score than the divergence-based variant. The performance gap is particularly notable in complex, expert-level datasets such as `Pen-Expert` and `Hammer-Expert`, suggesting that our method is more effective at modeling intricate, high-skill behaviors.

This trend of robust outperformance continues on the more diverse OGBench suite (Figure 4b), which includes challenging locomotion and navigation tasks. In environments like `Humanoid-Medium`, `Cube-Double`, and `Puzzle-4x4`, PQL again establishes a significant performance advantage. The consistent success across 22 distinct environments strongly indicates that our proposed regularization is a more effective and generalizable approach than the standard divergence-based alternative.

### E.2    ABLATION STUDY

Table 5: Ablation Study in Adroit environment. We ablate our method by removing the PDE regularizer (PQL-PDE) and the Beta-distributed time sampling (PQL-Beta)

| Algorithm | Pen | | | Door | | | Hammer | | | Relocate | | |
|---|---|---|---|---|---|---|---|---|---|---|---|---|
| | Human | Cloned | Expert | Human | Cloned | Expert | Human | Cloned | Expert | Human | Cloned | Expert |
| PQL-Beta (w/o Beta) | 75±8 | 80±10 | 140±9 | 2±2 | 1±1 | 100±5 | 2±2 | 10±8 | 128±6 | 0.5±0.2 | 0.2±0.2 | 105±3 |
| PQL-PDE (w/o PDE) | 53±12 | 72±11 | 143±4 | 0±0 | 2±1 | 102±2 | 1±1 | 11±9 | 123±3 | 0±0 | 0±0 | 106±1 |
| PQL | 82±3 | 87±6 | 148±7 | 4±1 | 3±0.8 | 107±2 | 4±1 | 14±6 | 134±1 | 0.8±0.1 | 0.5±0.1 | 110±1 |

Table 6: Ablation Study on selected OGBench tasks, grouped by environment.

| Algorithm | Antmaze | | Humanoid | | Antsoccer | Cube | | Scene | Puzzle | |
|---|---|---|---|---|---|---|---|---|---|---|
| | Large-nav | Giant-nav | Medium-nav | Large-nav | Arena-nav | Single-play | Double-play | Play | 3x3-Play | 4x4-Play |
| PQL-Beta (w/o Beta) | 75±10 | 4±2 | 18±9 | 7±0.6 | 37±13 | 83±1 | 34±2 | 70±2 | 15±3 | 13±1 |
| PQL-PDE (w/o PDE) | 77±28 | 5±3 | 20±11 | 8±1 | 40±31 | 97±3 | 35±7 | 77±8 | 17±4 | 12±2 |
| PQL | 84±7 | 7±1 | 30±10 | 9±1 | 41±15 | 100±1 | 41±4 | 80±3 | 19±3 | 16±1 |

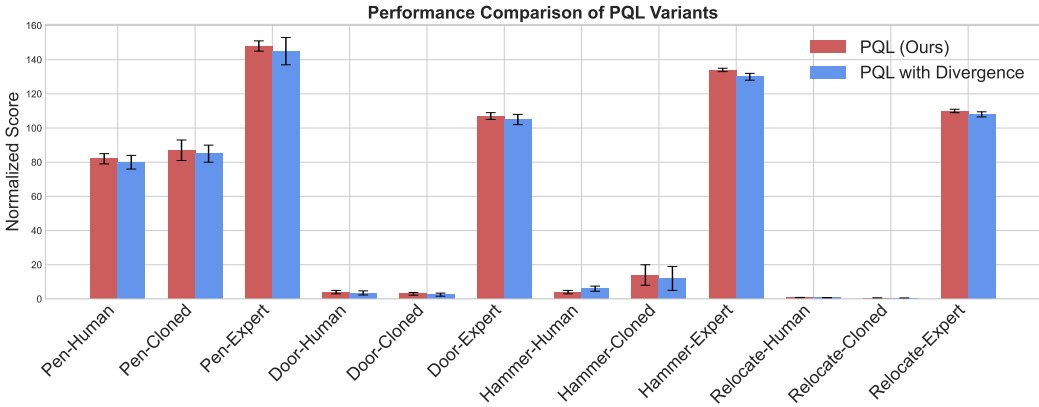

(a) Performance comparison across Adroit tasks.

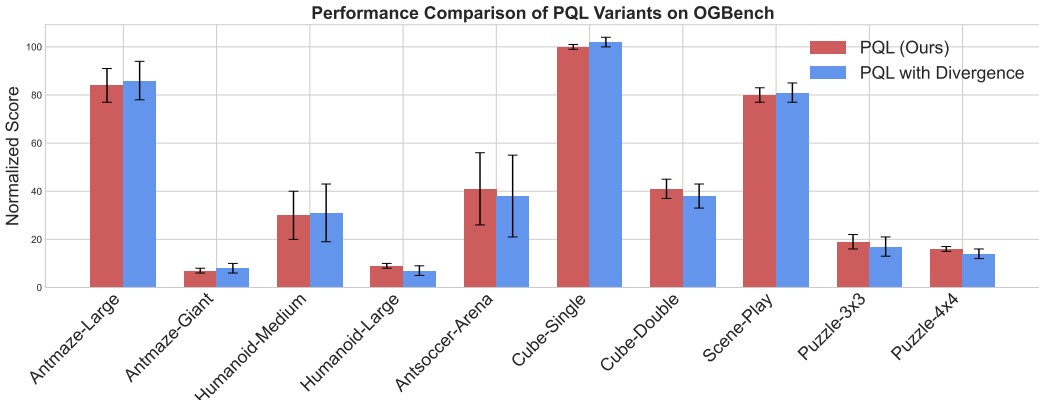

(b) Performance comparison across OGBench tasks.

Figure 4: Performance comparison of PQL (Ours) and PQL with Divergence across two benchmark suites: (a) Adroit and (b) OGBench. Error bars represent one standard deviation over multiple seeds.

To further validate the individual contributions of our proposed components, we conduct a thorough ablation study, with results presented in Table 5 for the Adroit suite and Table 6 for the OGBench suite. We systematically evaluate the impact of our two core contributions: the continuity-based PDE regularizer and the Beta-distributed time sampling strategy.

**Impact of the PDE Regularizer.** We first analyze the impact of our central contribution by comparing PQL against the **PQL-PDE (w/o PDE)** variant, which removes the regularizer. The results unequivocally demonstrate that the PDE constraint is critical for performance. Across the Adroit tasks (Table 5), removing the regularizer causes a substantial performance collapse, especially on the challenging human and cloned datasets such as `Pen-Human` (82 vs. 53) and `Hammer-Cloned` (14 vs. 11). This trend is mirrored in the OGBench results (Table 6), where the performance gap remains significant in complex tasks like `Antmaze-Large-nav` (84 vs. 77) and `Humanoid-Medium` (30 vs. 20). This provides strong empirical evidence for our main hypothesis: that leaving the intermediate probability path unconstrained is a key performance bottleneck in flow-matching policies, and that enforcing physical consistency via our PDE regularizer directly and effectively addresses this limitation.

**Impact of Beta-distributed Time Sampling.** Next, we investigate the importance of our proposed time sampling strategy by evaluating the **PQL-Beta (w/o Beta)** variant, which reverts to standard uniform sampling. The results in both tables show that this change also leads to a consistent degradation in performance. For instance, in `Pen-Cloned` (87 vs. 80) and `Cube-Single` (100 vs.

83), the absence of Beta sampling yields significantly lower scores. This validates our second claim: that naively optimizing the PDE-constrained objective is challenging. The superior performance of our full PQL model demonstrates that Beta-distributed sampling, by concentrating updates in the most critical mid-range of the generative trajectory where data and noise are maximally entangled, provides a more stable and effective training signal. This tailored optimization strategy is crucial for unlocking the full potential of the PDE constraint. In summary, our ablation studies confirm that both the PDE regularizer and the Beta-distributed time sampling are integral to PQL's success. The removal of either component leads to a significant performance drop, affirming that our method's strong results stem from the synergy between a principled physical constraint and an effective, tailored optimization strategy.

**Discussion** It is instructive to contextualize our contributions with respect to recent state-of-the-art flow-matching methods such as FQL. While highly effective, our experiments suggest that unconstrained flow-matching policies can exhibit significant performance variance on complex tasks. For instance, our `PQL-PDE (w/o PDE)` baseline, which is analogous to FQL, demonstrates high variance in challenging environments like `Antmaze-Large-nav` and `Antsoccer-Arena-nav`, as shown in Table 6 and Table 3. In contrast, our full PQL model not only achieves a higher mean score but also substantially reduces this variance, highlighting the stabilizing effect of our PDE regularizer. Furthermore, this analysis clarifies the nuanced role of our Beta-distributed time sampling strategy. Prior work Park et al. (2025b) suggest that such sampling schemes offer little general benefit for standard flow-matching objectives, a finding that aligns with the moderate performance drop of our `PQL-Beta (w/o Beta)` ablation. However, in our work, the view would be different: the Beta distribution is not proposed as a universal performance enhancer, but as a specific solution to the more challenging optimization landscape introduced by our PDE regularizer. By enforcing physical consistency, the regularizer creates a more structured but complex learning problem. The Beta sampling strategy is essential to effectively navigate this landscape by focusing the training on the most informative parts of the generative trajectory. Therefore, the necessity of our approach lies in the synergy of its components: the PDE regularizer imposes a crucial physical constraint, and the Beta sampling makes this principled constraint amenable to effective optimization.

### E.3 HYPER-PARAMETERS

**Beta Sampling $\alpha$** To investigate the impact of our proposed Beta-distributed time sampling strategy, we conduct a comprehensive sensitivity analysis on the key hyperparameter, $\alpha$, which controls the shape of the sampling distribution. The results across all three benchmark suites—Adroit, OGBench, and Gym-MuJoCo—are summarized in the heatmaps in Figure 5. The figure visualizes the normalized score for $\alpha$ values ranging from 1.0 (equivalent to uniform sampling) to 5.0. The results present a clear and consistent trend across all 31 distinct environments. Optimal performance is almost universally achieved when $\alpha$ is set to either 2.0 or 3.0, as indicated by the bolded scores and highlighted cells in both heatmaps. This finding strongly confirms our central hypothesis: concentrating the sampling of the time variable $t$ in the critical mid-range of the generative trajectory—where data and noise are most entangled—provides a more effective and generalizable learning signal for the PDE-constrained objective. The consistency of this result across tasks with vastly different dynamics, from complex manipulation (Adroit) to navigation (OGBench) and locomotion (Gym-MuJoCo), underscores the robustness of our approach.

Furthermore, the analysis reveals a significant performance degradation at the extremes. The notably lower scores at $\alpha = 1.0$ across nearly every task provide strong, widespread evidence against uniform time sampling, validating a core claim of our work. Similarly, a very high $\alpha$ (e.g., 5.0), which creates an overly narrow sampling peak, also tends to degrade performance. Based on this comprehensive analysis, we selected an $\alpha$ value of 3.0 for all main experiments, as it provided the most consistent and high-performing results across the full suite of environments.

**PDE Temperature $\lambda$** In addition to the time-sampling distribution, a second critical hyperparameter is the PDE temperature, $\lambda$, which serves as the weighting coefficient for our continuity-based regularization term in the final loss function. This parameter governs the fundamental trade-off between two competing objectives: the fidelity of the terminal action distribution and the physical consistency of the generative path. An overly large $\lambda$ could force the model to prioritize a smooth path at the expense of accurately capturing the complexity of the target action distribution, resulting

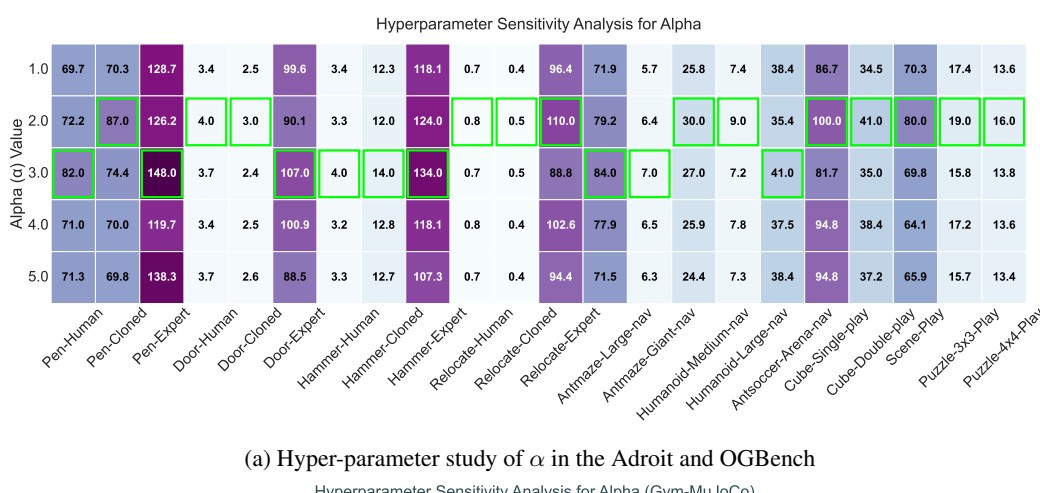

(a) Hyper-parameter study of $\alpha$ in the Adroit and OGBench

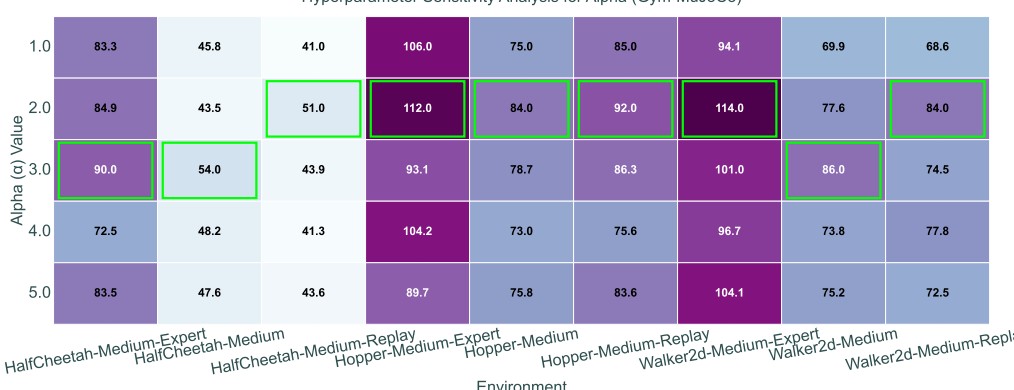

(b) Hyper-parameter study of $\alpha$ in the MuJoCo

Figure 5: Comprehensive hyper-parameter study of the Beta-distribution parameter $\alpha$ across all three benchmark suites. (a) shows results on Adroit and OGBench tasks, while (b) shows results on Gym-MuJoCo tasks.

in a suboptimal policy. Intuitively, we hypothesize that a small, non-zero value for $\lambda$ is required to balance these objectives effectively. To empirically determine this balance, we perform a detailed sensitivity analysis on $\lambda$. Figure 6 presents the results of this study across all 31 environments, split into two heatmaps for clarity: (a) for Gym-MuJoCo and (b) for the combined Adroit and OGBench suites. We tested $\lambda$ values spanning several orders of magnitude to thoroughly map the performance landscape. The results provide a clear and remarkably consistent conclusion across all three distinct benchmarks. As hypothesized, the performance of PQL is highly sensitive to the value of $\lambda$, and there is a distinct optimal range. The highest scores are almost universally achieved for small values of $\lambda$, specifically within the range of $[0.01, 0.02]$, as highlighted by the cells in both figures. This demonstrates that only a small weighting is necessary for the PDE regularizer to effectively stabilize the generative path without interfering with the primary policy optimization objective. The consistency of this finding across a wide array of task complexities and dynamics speaks to the robustness of our proposed method. Furthermore, the heatmaps clearly illustrate the trade-off we aimed to balance. When $\lambda$ is too large (e.g., $\lambda = 1.0$), performance consistently and significantly degrades across every single environment. This result strongly supports our claim that overly prioritizing the physical consistency of the flow can harm the final policy's performance. Conversely, the strong results in the low-$\lambda$ regime confirm that the regularizer is a critical component for success. Based on this comprehensive study, we selected $\lambda = 0.015$ for all main experiments, as it represents the most consistent point of optimal performance across the full set of environments.

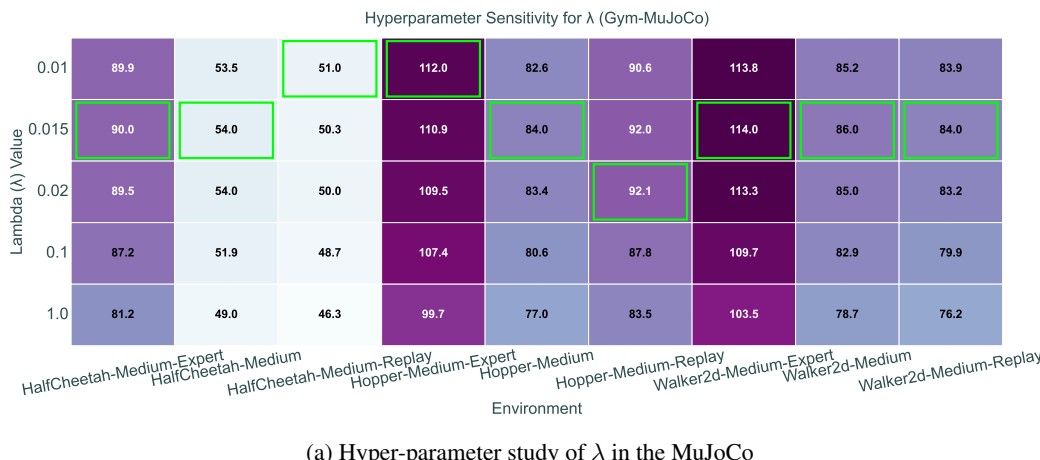

(a) Hyper-parameter study of $\lambda$ in the MuJoCo

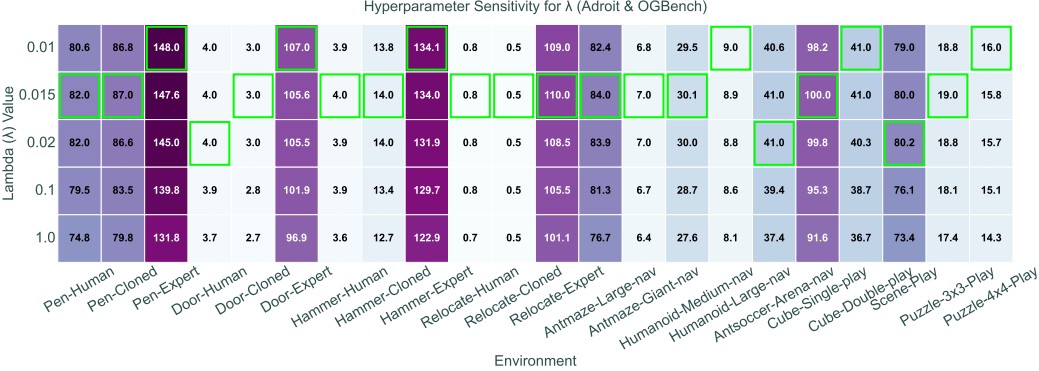

(b) Hyper-parameter study of $\lambda$ in the Adroit and OGBench

Figure 6: Comprehensive hyper-parameter study of the PDE parameter $\lambda$ across all three benchmark suites. (a) shows results on Gym-MuJoCo tasks, while (b) shows results on Adroit and OGBench tasks.

## F  Training and Implementation

The overall training algorithm can be found in Algorithm 1.

We also provide the complete list of hyperparameters in Table 7 for reproducibility.

Table 7: Hyperparameters for PQL.

| Hyperparameter | Value |
|---|---|
| Learning rate | 0.0003 |
| Optimizer | AdamW ((Loshchilov & Hutter, 2017)) |
| Gradient steps | 1000000 (default), 500000 (D4RL, pixel-based OGBench) |
| Minibatch size | 256 |
| MLP dimensions | [512, 512, 512, 512] |
| Nonlinearity | GELU ((Hendrycks & Gimpel, 2016)) |
| Target network smoothing coefficient | 0.005 |
| Discount factor $\gamma$ | 0.99 (default), 0.995 (antmaze-giant, humanoidmaze, antsoccer) |
| Flow time sampling distribution | $\text{Beta}(\alpha, \alpha)$ with weighted sampling |
| Beta Parameter | $\alpha = 2$ or $\alpha = 3$, please refer to Appendix E.3. |
| PDE coefficient $\lambda$ | 0.015 |

---

**Algorithm 1** PDE-Regularized and Beta-Weighted Flow Matching for Offline RL

---

**Require:** Offline dataset $\mathcal{D} = \{(s, a, r, s')\}$, parameters $\theta$ for actor (velocity field $u_\theta$), parameters $\phi$ for critic $Q_\phi$, regularization weight $\lambda_{\text{jac}}$, Beta distribution $\pi_{\alpha,\beta}$.

1: Initialize $\theta, \phi$ randomly.
2: **for** each training iteration **do**
3:     Sample minibatch $\{(s, a, r, s')\} \subset \mathcal{D}$.
4:     Sample noise actions $a_0 \sim \mathcal{N}(0, I)$.
5:     Sample timesteps $t \sim \pi_{\alpha,\beta}$.
6:     Construct interpolants $a_t = (1 - t)a_0 + ta$.
7:     Compute velocity targets $v = a - a_0$.
8:     Compute predicted velocity $u_\theta(s, a_t, t)$.
9:     **Flow matching loss with Beta sampling:**

$$\mathcal{L}_\pi(\theta) = \mathbb{E}\left[w_t^\pi \left\| u_\theta(s, a_t, t) - (a_1 - a_0) \right\|^2\right], \quad w_t^\pi = \frac{t}{1-t}\pi(t).$$

10:     **Jacobian regularizer:**

$$\mathcal{L}_{\text{PDE}}(\theta) = \lambda \, \mathbb{E}\left[\|\nabla_a u_\theta(s, a_t, t)\|_F^2\right].$$

    (Estimated with Hutchinson's trick.)
11:     **Q-loss:** Sample actions $\hat{a} \sim u_\theta$, compute

$$\mathcal{L}_Q(\phi) = \mathbb{E}\left[\left(Q_\phi(s, \hat{a}) - (r + \gamma \max_{a'} Q_\phi(s', a'))\right)^2\right].$$

12:     **Actor objective:**

$$\mathcal{L}_{\text{actor}}(\theta) = \mathcal{L}_\pi(\theta) + \lambda \mathcal{L}_{\text{PDE}}(\theta) + \mathbb{E}[Q_\phi(s, a_1')] + \mathcal{L}_{distill}.$$

13:     Update actor parameters: $\theta \leftarrow \theta - \eta_\theta \nabla_\theta \mathcal{L}_{\text{actor}}$.
14:     Update critic parameters: $\phi \leftarrow \phi - \eta_\phi \nabla_\phi \mathcal{L}_Q$.
15: **end for**

---

