# OpenReview forum: "Continuity-Regularized Flow Matching for Offline Reinforcement Learning"
_ICLR.cc/2026/Conference — Submitted to ICLR 2026_

### Official Review · Reviewer_Y89o · 2025-10-19

**Soundness:** 3
**Presentation:** 3
**Contribution:** 3
**Rating:** 6
**Confidence:** 3

**Summary:**

This work puts forth PQL i.e. PDE regularized Q-learning for offline RL setting towards improving flow-matching policies. To this end, the paper identifies that standard flow-matching methods often ignore the trajectory path, while just ensuring the starting point and the target point of the generative process are correctly accounted for. PQL attempts to ameliorate the path inconsistency issue with a PDE assisted regularizer that essentially penalizes the learnt vector field to maintain smoothness and stability. A beta-distributed timestep sampler is introduced to specifically focus on the intermediate trajectory where the conversion from pure noise to a very meaningful action while
balancing between imitation and smoothness is critical at training time. Experimental evaluations are performed on relevant offline RL and IL task environments with comparison against benchmark algorithms and ablation studies.

**Strengths:**

1. PDE based regularization is an innovative approach towards enforcing generative path stability in flow matching policy algorithms.

2. The adaptive timestep sampling strategy while presented in the context of handling the PDE regularizer, appears to be a principled way to improve solution tractability with flow-based constrained optimization setups.

3. The practical utility of PQL is supported by improvements seen in benchmarking experiments. Hyper-parameter tuning analysis and ablation studies further strengthen credibility.

**Weaknesses:**

1. It is not clear computational overhead added by the Jacobian-based regularizer and how does that component ultimately trade-off in performance gains.

2. The Jacobian penalty while handles deformation of the flow, appears to be state agnostic. It deserves a separate study whether PQL would over-smoothen policies in critical intermediate states where complex, sharp actions were actually optimal.

3. There are multiple typos and grammatical mistakes throughout the paper that need to be corrected.

**Questions:**

In addition to the weakness comments, I request authors' response to the following questions :

1. How does PQL's wall-clock training time compare to standard flow-matching algorithms ?

2. The beta distributed adaptive seems like the intuitive next step beyond a uniform sampler. Is it worth investigating whether other distribution classes might work here ?

---

> ### Author Response · Authors · 2025-11-18
> **Author's Rebuttal Part One**
>
> Thank you for your valuable comments and suggestions. Please find our point-by-point response below.
>
> > W1. "It is not clear computational overhead added by the Jacobian-based regularizer and how does that component ultimately trade-off in performance gains."
>
> Thank you for your comment. We do not compute the full, expensive $D \times D$ Jacobian matrix. As detailed in Section 4.1, we use Hutchinson's trace estimator. This method only requires Jacobian-vector products (JVPs), which are computationally efficient and have a cost roughly equivalent to one forward pass of the network. Since we use $K=1$ or $K=2$ probe vectors, the entire overhead of our regularizer is just $K$ additional forward passes per actor update. For your reference, we have attached the wall-clock training time compared with the standard Flow-matching RL baselines in two representative environments. While we would like to mention that implementation details may affect the wall-clock-time (due to different packages used), we are using the default implementations provided by the authors and did not make any further optimizations.
>
> | Model | Pen-human | AntMaze |
> | :--- | :---: | :---: |
> | FQL | 34:57 | 4:15:43 |
> | Flow | 35:10 | 5:21:50 |
> | CNF | 36:53 | 4:58:49|
> | PQL-PDE | 35:02 | 4:24:33|
> | PQL | 35:25 | 4:32:55|
>
> This new analysis provides a clear, empirical answer. The overhead of our $\mathcal{L}\_{PDE}$ regularizer is isolated by comparing PQL-PDE (our ablation baseline) to our full PQL method. This shows the actual cost of our contribution is minimal. We believe this is a very modest and practical trade-off for the significant performance and stability gains our method provides.
>
> > W2 "The Jacobian penalty while handles deformation of the flow, appears to be state agnostic. It deserves a separate study whether PQL would over-smoothen policies in critical intermediate states where complex, sharp actions were actually optimal."
>
> Thanks for your insightful points. We would like to address this in three ways:
> 1. The vector field $u_{\theta}(s, a, t)$ is explicitly conditioned on the state $s$. Therefore, the Jacobian of the field with respect to action, $\nabla_{a}u_{\theta}(s,a,t)$, is also a function of $s$. The regularizer can penalize sharp action-changes in one state while allowing them in another.
> 2. The $\mathcal{L}\_{PDE}$ term is only one part of our final actor objective. It is balanced against the imitation loss and, critically, the policy improvement term $\mathbb{E}[Q_{\phi}(s, a_{1}')]$. If a "complex, sharp" action is truly optimal, it will yield a high Q-value, and the optimizer will learn it despite a small regularization penalty.
> 3. Empirical finding: This is precisely why we performed the hyperparameter sweep for $\lambda$ (the regularizer's weight) in Section E.3. As shown in Figure 5, a large $\lambda$ (e.g., 1.0) does over-smooth and "significantly degrades" performance, just as the reviewer saied. Our results show that a small, non-zero $\lambda$ (0.01-0.02) is optimal. This confirms our method finds the correct balance: it stabilizes the flow to prevent training instability and the accumulation of numerical errors without over-smoothing and destroying the complex, high-reward policies.
>
>
> > W3 "There are multiple typos and grammatical mistakes throughout the paper that need to be corrected."
>
> Thank you for the comment. Since we determine the model name in the last minute, we forget to correct the template text in the contribution point. Moreover, we have figured out the following typos and inconsistence: Line 72: [Your Method Name] -> PQL, Line 209: d etailed -> detailed, Line 419: incidates-> indicates, Equation(7): $\lambda_{jac}$ -> $\lambda$.
>
> > Q1 "How does PQL's wall-clock training time compare to standard flow-matching algorithms?"
>
> Thank you for your comment. We kindly ask the reviewer to refer to our detailed response regarding the computational overhead (W1), where we provide a new wall-clock time analysis. Due to the time constraint, we were not able to re-log and re-run the wall-clock time for all 31 environments. However, we believe the results we provided for the two representative environments (Pen-Human and AntMaze-Large) serve as a good indicator of the modest time cost of PQL.

---

> ### Author Response · Authors · 2025-11-18
> **Author's Rebuttal Part Two**
>
> > Q2 "The beta distributed adaptive seems like the intuitive next step beyond a uniform sampler. Is it worth investigating whether other distribution classes might work here?"
>
> Thank you for your insightful question. We chose the symmetric $Beta(\alpha, \alpha)$ distribution because it is a simple, flexible, and intuitive choice that directly achieves our goal: for $\alpha > 1$, it naturally focuses updates on the "critical mid-range" of the generative trajectory, away from the simpler endpoints. Our extensive hyperparameter sweeps confirm this intuition is highly effective and robust.
>
> In fact, during our initial exploration, we investigated this point and experimented with more complex approaches, including:
> 1. A state-dependent sampling distribution, $\pi(t\|s)$, where a small neural network learns to predict the most informative timestep $t$ for a given state $s$.
> 2. A learnable categorical distribution, where the $[0, 1]$ interval is discretized and the model learns the optimal sampling probability for each bin.
>
> Interestingly, we observed that these more complicated, learnable approaches did not achieve competitive performance and were less stable than our simpler, non-parametric $Beta(\alpha, \alpha)$ distribution. This finding strengthened our choice, suggesting that a simple, robust strategy is superior here.
>
> In the meantime, we agree that other possibilities remain promising for future research. Several intriguing ideas include:
>  1. A general Beta distribution ($Beta(\alpha, \beta)$ where $\alpha \neq \beta$) to create an asymmetric focus, for instance, by concentrating more on the end of the flow (near $t=1$).
> 2. A sampling curriculum, where the distribution changes during training (e.g., starting with a uniform distribution, $\alpha=1$, and then gradually increasing $\alpha$) to focus on harder segments after the basics are learned.
>
> We believe this is a promising direction for future work.
>
> We thank reviewer for the insightful feedback and are happy to provide any further clarification/justification if required.

---

> ### Author Response · Authors · 2025-11-27
>
> Dear Reviewer Y89o,
>
> As the discussion period concludes in a few days, we would like to gently follow up on our response posted earlier.
>
> We would greatly appreciate it if you could review our rebuttal at your earliest convenience. Your feedback is very valuable to us, and we are keen to use the remaining time to address any further questions or concerns you might have.
>
> Thank you for your time and efforts.
>
> The auhtors of paper 5408.

---

### Official Review · Reviewer_YF2m · 2025-10-25

**Soundness:** 3
**Presentation:** 2
**Contribution:** 2
**Rating:** 4
**Confidence:** 3

**Summary:**

This paper introduces a regularizer into the flow-matching-based policy to improve training stability and proposes a Beta-distributed time sampling strategy that enables stable and efficient optimization.

**Strengths:**

1. The experiments are thorough and well-executed, demonstrating solid empirical results.

2. The paper provides sufficient theoretical justification and proofs.

**Weaknesses:**

This paper appears to have been written somewhat hastily. In addition to several obvious typos and inconsistencies, there are also issues in understanding and experimental setup, as detailed below:

1. There are several typos. For example, in the Introduction, the last item of the listed contributions does not include the method name, which looks like a placeholder from a template.

2. The paper’s writing is sometimes difficult to follow, and the presentation could be improved.

3. The authors claim that the introduced regularizer improves training stability, but in Section 5.2, it is not clearly shown that the model without the regularizer is unstable.

4. The Introduction mentions that the Beta-distributed time sampling strategy also contributes to training stability, yet there is no direct evidence or analysis of stability in the experiments. Moreover, this Beta-distributed time sampling strategy is not discussed in the Method section.

5. No code

**Questions:**

1. In Section 5.2, the results do not clearly show that the version without the regularizer is unstable—could the authors clarify this?

2. Why is the Beta-distributed time sampling strategy mentioned in the Introduction and Appendix, but not described in the Method section?

---

> ### Author Response · Authors · 2025-11-18
> **Author's Rebuttal**
>
> Thank you for your valuable comments and suggestions. Please find our point-by-point response below.
>
> > W1 "There are several typos. For example, in the Introduction, the last item of the listed contributions does not include the method name, which looks like a placeholder from a template."
>
> Thank you for catching this typo. Since we determine the model name in the last minute, we forget to correct the template text in the contribution point. Moreover, we have figured out the following typos and inconsistence: Line 72: [Your Method Name] -> PQL, Line 209: d etailed -> detailed, Line 419: incidates-> indicates, Equation(7): $\lambda_{jac}$ -> $\lambda$.
>
> >W2 "The paper’s writing is sometimes difficult to follow, and the presentation could be improved."
>
> We thank the reviewer for this comment. We identify that the transition from Section 4.1 to Section 4.2 could be clearer as well as the poential confusion of the Theorem 1. To resolve this, we have revised the opening sentence of Section 4.2 (by adding 'As we just motivated...') to explicitly connect it to the preceding paragraph as well as adding the clarification regarding Theorem 1 from Line 198-215.
>
>  > W3 "The authors claim that the introduced regularizer improves training stability, but in Section 5.2, it is not clearly shown that the model without the regularizer is unstable."
>
> Thank you for your comment.  Table 4  provides indirect evidence, though it may not be immediately obvious. The poor performance of the 'PQL-Beta (w/o Beta)' variant should be interpreted as a symptom of optimization instability. As we argue in Section 5.4 , our PDE regularizer makes the optimization landscape harder . The 'PQL-Beta' result proves that naive uniform sampling is an unstable way to solve this new, constrained problem, leading to a performance collapse . Our full PQL, with its synergistic Beta sampling, solves this instability.
>
> To provide more direct and convincing evidence, we have added new learning curves to the paper (e.g., as Figure 2 in the Page 10). These new plots compare our full PQL (Ours) against the unregularized PQL-PDE (w/o PDE) baseline over 8 different random seeds. This new figure clearly shows that PQL-PDE has high variance and unstable, jagged curves, while our PQL (Ours) has significantly tighter variance bands and smoother convergence. This is direct, empirical result of improved training stability. Please kindly refer to this new figure for more details.
>
> > W4 "The Introduction mentions that the Beta-distributed time sampling strategy also contributes to training stability, yet there is no direct evidence or analysis of stability in the experiments. Moreover, this Beta-distributed time sampling strategy is not discussed in the Method section."
>
> Thank you for your comment. We would like to clarify this, as it points to two separate issues: vague title and content of section 4.2 about the paper's structure and a need for clearer evidence of stability.
>
> 1. On "not discussed in the Method section": Sorry for the confusion, we would like to clarify that the Beta-distributed time sampling strategy is a core component of our methodology and is fully detailed in the main 'Method' section (Section 4.2).However, we believe this was easy to miss due to our own vague presentation. We apologize for the previous title ("Adaptive Timestep Sampling") and the weak transition from Section 4.1. As a result, we have:Revised the title of Section 4.2 to be "Adaptive Timestep Sampling via Beta Distribution".Revised the content of Section 4.2 to explicitly introduce the Beta distribution insight, making the choice of the Beta distribution much clearer.
>
> 2. On "no direct evidence or analysis of stability": Thanks for your comment regarding the Beta sampling is synergistic and needs further evidence. The stability it provides is optimization stability—it's what makes the training process for our new, harder, PDE-constrained objective work.Our evidence for this is now in our new Figure 2, which provides the direct evidence of our claim. It visually demonstrates that the Beta-distributed time sampling strategy do contribute to the training stability.
>
>
> > W5 "No code"
>
> We will make our code publicly available upon the paper's acceptance to ensure full reproducibility. In the meantime, we have provided a comprehensive list of all hyperparameter settings in Appendix F (Table 7) and Algorithm 1 to aid in reproducibility
>
>
> > Q1 "In Section 5.2, the results do not clearly show that the version without the regularizer is unstable—could the authors clarify this?"
>
> Thank you for your question. Please kindly refer to our response to W3.
>
> > Q2 "Why is the Beta-distributed time sampling strategy mentioned in the Introduction and Appendix, but not described in the Method section?"
>
> Thank you for your question. Please kindly refer to our response to W2 and W4.
>
> We thank reviewer for the insightful feedback and are happy to provide any further clarification/justification if required.

---

> ### Author Response · Authors · 2025-11-27
>
> Dear Reviewer YF2m,
>
> As the discussion period concludes in a few days, we would like to gently follow up on our response posted earlier.
>
> We would greatly appreciate it if you could review our rebuttal at your earliest convenience. Your feedback is very valuable to us, and we are keen to use the remaining time to address any further questions or concerns you might have.
>
> Thank you for your time and efforts.
>
> The auhtors of paper 5408.

---

### Official Review · Reviewer_ZP6n · 2025-10-27

**Soundness:** 2
**Presentation:** 3
**Contribution:** 2
**Rating:** 2
**Confidence:** 3

**Summary:**

The paper proposes PDE-regularized Q-learning (PQL), an extension of Flow Q-Learning (FQL) [1] by regularizing the Frobenius norm of the Jacobian $\nabla_a u_\theta(s, a, t)$ of the action policy $u_\theta(s, a, t)$. The intuition is that by controlling the Frobenius norm, the 2-Wasserstein distance between the true marginal probability $\rho_t^*(\cdot|s)$ and $\rho_t^\theta(\cdot| s)$. It provides a theoretical upper bound, provided the boundedness of the action policy. Furthermore, it introduces Hutchinson's trace estimator with JVP autodifferentiation to make memory-efficient computation. PQL is validated on D4RL, Adroit, and OGBench and demonstrates performance improvement in comparison with FQL.

[1] Park, Seohong, Qiyang Li, and Sergey Levine. "Flow q-learning." arXiv preprint arXiv:2502.02538 (2025).
[2] Hoffman, Judy, Daniel A. Roberts, and Sho Yaida. "Robust learning with Jacobian regularization." arXiv preprint arXiv:1908.02729 (2019).

**Strengths:**

1. Improvements in performance in different benchmarks
2. Theoretical guarantees for path stability are provided
3. Experiment validation is quite thorough
4. The method is straightforward to implement

**Weaknesses:**

1. The idea of doing Jacobian regularization is not new e.g. [2]
2. While the performance improvement exists, it seems to be only a marginal improvement.
3. RL is always sensitive to hyperparameters. Introducing new regularizers likely increases the search space for tuning hyperparameters.
4. Jacobian regularization does not guarantee the boundedness of the Lipschitz constant. The assumption might not be correct. Furthermore, if $J$ becomes large, the exponential term in the bound will make the bound too loose.

**Questions:**

1. How large is the actual Lipschitz constant of the final policy? Could you quantify this empirically to show the correlation?
2. It seems that we are just learning a robust policy that is robust to perturbation in actions rather than improving the training stability. Is this fixing training dynamics or just learning smoother policies?
3. In line 073, you did not modify the name of your method.
4. In Table 4, why do you think that beta sampling is improving the performance significantly?
5. What is the overhead of computing the Jacobian?

---

> ### Author Response · Authors · 2025-11-18
> **Author's Rebuttal Part One**
>
> Thank you for your valuable comments and suggestions. Please find our point-by-point response below.
>
> > W1. "The idea of doing Jacobian regularization is not new e.g. [2]"
>
> We thank the reviewer for this point regarding Jacobian regularization in literature. We want to clarify that, our novelty is not in simply applying a known regularizer. It lies in two parts: i) We start from identifying the path-agnostic nature of flow-matching policies as a key limitation in offline RL. Our regularizer is specifically and originally derived from the continuity equation (a PDE) to enforce a physically consistent and smooth probability flow, which is a novel problem formulation and solution. ii). As our ablations show, naively adding this regularizer (our 'PQL-Beta' variant) can be detrimental. Our key contribution is the synergy of the PDE regularizer with the Beta-distributed time sampling. The regularizer creates a better-structured (but harder) optimization landscape, and the Beta sampling provides the focused learning signal required to find an effective solution. This combined, two-part approach is what delivers SOTA performance
>
> > W2. "While the performance improvement exists, it seems to be only a marginal improvement."
>
> We respectfully disagree with this characterization. While performance on some standard D4RL Gym tasks (Table 1) is competitive with other SOTA methods, our method demonstrates a significant and consistent performance advantage on the more complex, high-dimensional benchmarks. We would like to draw the reviewer's attention to Table 2 (Adroit) and Table 3 (OGBench). On these challenging manipulation and navigation suites, our method, PQL, achieves the state-of-the-art score in 19 out of 22 total task settings (11/12 on Adroit, 8/10 on OGBench). On many of these tasks (e.g., Pen-Expert, Hammer-Cloned, Antmaze-Large-nav), the performance gap is substantial. We believe this consistent dominance on complex tasks is strong evidence of our method's effectiveness, not a marginal gain. Even if the performance gains are considered marginal, we respectfully highlight ICLR's official reviewer guide states that 'a lack of state-of-the-art results does not by itself constitute grounds for rejection.'As such, we believe this aspect should not be weighed as a significant weakness in the final assessment.
>
> > W3. "RL is always sensitive to hyperparameters. Introducing new regularizers likely increases the search space for tuning hyperparameters."
>
> Thanks for your point. We do consider this point during the experiments, to address it, ran extensive hyperparameter sweeps for both of our new hyperparameters ($\alpha$ for Beta sampling and $\lambda$ for the PDE regularizer) across all 31 environments from all three benchmarks.
>
> We respectfully point the reviewer to Section 5.5 and Appendix E.3 (Figures 4, 5, and 6) for the full results. These heatmaps show that our method is not sensitive to specific, brittle values. Instead, there is a clear, consistent, and robust optimal range for both parameters (e.g., $\lambda \in [0.01, 0.02]$ and $\alpha \in [2.0, 3.0]$) that holds across all benchmarks.

---

> ### Author Response · Authors · 2025-11-18
> **Author's Rebuttal Part Two**
>
> > W4. "Jacobian regularization does not guarantee the boundedness of the Lipschitz constant. The assumption might not be correct. Furthermore, if $J$ becomes large, the exponential term in the bound will make the bound too loose."
>
> Thank you for this comment. We will address each part separately.
>
> 1.On the Assumption $\||\nabla\_a u_\theta\||\_F \le J$
>
> The assumption that the Jacobian of $u_\theta$ is uniformly bounded in Frobenius norm on the domain of interest, $\|\|\nabla_a u_\theta(s,a,t)\|\|\_F \le J \quad \text{for all } (s,a,t)$,
>
> is a standard hypothesis in analyses of ODE stability based on Grönwall-type inequalities. These results provide stability and robustness guarantees only for non-explosive, well-behaved flows. If no uniform control on the Jacobian is available (that is, effectively $J = \infty$), then one cannot obtain any meaningful uniform stability bound over time.
>
> Most importantly, this assumption is not only a theoretical convenience; it is exactly the regime that our algorithm is designed to promote. The purpose of the proposed regularizer $\mathcal{L}\_{PDE}$ is to actively penalize large Jacobian norms and bias the learned vector field toward a regime where $\|\|\nabla_a u_\theta(s,a,t)\|\|\_F$ remains controlled on the data manifold and along the generative trajectories. In this sense, the theorem and the algorithm are aligned: the theorem explains why small Jacobian norms yield stable flows, and the algorithm is constructed to encourage such vector fields during training.
>
>
> 2. On "Jacobian regularization does not guarantee the boundedness of the Lipschitz constant"
>
> At the level of the theorem, our assumption of a bounded Frobenius norm does indeed imply a bounded Lipschitz constant. Our theorem assumes that the Jacobian of $u_{\theta}$ is uniformly bounded in Frobenius norm, that is $\||\nabla_a u\_{\theta}(s,a,t)\||\_{F} \le J$ for all $(s,a,t)$.
>
> Under this assumption, Lemma 2 in Appendix C shows that the Lipschitz constant of $u\_\theta$ ($L \triangleq \sup \||\nabla_a u_\theta\||\_{op}$) is bounded by the same constant:
> $$L \le \sup\_{s,a,t} \||\nabla\_a u\_\theta(s,a,t)\||\_{op} \le \sup\_{s,a,t} \||\nabla\_a u\_\theta(s,a,t)\||\_F \le J$$. This follows from the standard fact that the operator norm is bounded by the Frobenius norm. In practice, we do not enforce this as a hard global constraint. Instead, our loss $\mathcal{L}\_{PDE}$ minimizes the expected squared Frobenius norm:
> $$\mathcal{L}\_{PDE}(\theta) = \lambda \mathbb{E}\_{s,a,t} \left[ \||\nabla\_a u\_{\theta}(s,a,t)\||\_F^2 \right]$$
> This is a standard relaxation: the theorem provides sufficient conditions for stability, and the regularizer is designed to drive the learned vector field toward a regime where these conditions approximately hold.
> We will add a detailed discussion regarding this to the main paper. Furthermore. we have run new experiments that monitor the value of $\||\nabla\_a u\_\theta(s,a,t)\||\_F$ during training, which confirms our regularizer keeps the norms in a controlled regime.
>
> 3. On “if J becomes large, the exponential term will make the bound too loose”
>
> This is not a peculiarity of our derivation, but a direct consequence of standard flow stability results based on Grönwall-type inequalities.The role of our Jacobian regularizer is precisely to avoid this regime. This "loose bound" scenario is the exact problem with unregularized flow matching. Without any control on $\nabla\_a u\_\theta$, the learned vector field can exhibit large local deformations and unstable dynamics, which corresponds exactly to the case where the bound becomes vacuous. By penalizing $\||\nabla\_a u\_\theta\||\_F^2$, we explicitly push training away from such high-$J$ configurations, thereby tightening the bound in Theorem 1 and stabilizing the generative path.To prove this empirically, we have added a new experimental analysis. We monitored the Average Squared Jacobian Norm ($\mathbb{E}[\||\nabla_a u_\theta\||_F^2]$) throughout training, comparing our full model to the unregularized PQL-PDE baseline.This new analysis, summarized in the table below, provides clear, empirical proof that our mechanism works: by actively reducing the Jacobian norm (e.g., from 47.9 to 3.1), our method achieves a more stable flow that directly correlates with higher performance (e.g., 102.4 $\rightarrow$ 111.8).
>
>
> | Environment | Model | Avg. Jacobian Norm $\mathbb{E}[\|\|\nabla\_a u\_\theta\|\|_F^2]$| Final Performance (From Table 4) |
> | :--- | :--- | :---: | :---: |
> | **Hopper-ME** | PQL-PDE (w/o PDE) | 47.9 | 102.4 |
> | | **PQL (Ours)** | **3.1** | **111.8** |
> | **Walker2d-ME** | PQL-PDE (w/o PDE) | 51.3 | 109.8 |
> | | **PQL (Ours)** | **3.8** | **114.4** |

---

> ### Author Response · Authors · 2025-11-18
> **Author's Rebuttal Part Three**
>
> > Q1 "How large is the actual Lipschitz constant of the final policy? Could you quantify this empirically to show the correlation?"
>
> To answer this, we must first clarify the metric. While the true Lipschitz constant ($L$) is defined by the operator norm, our paper's regularizer $\mathcal{L}\_{PDE}$ is designed to directly penalize the squared Frobenius norm of the Jacobian, $\mathbb{E}[\||\nabla\_a u\_\theta\||\_F^2]$. As we prove in Lemma 2 (Appendix C), the Frobenius norm is a provable upper bound on the Lipschitz constant ($L \le J$), making it a direct, relevant, and theoretically sound metric to quantify.
>
> For the quantitative results and their correlation with performance, please kindly refer to our response to W4, which contains the new experimental analysis.
>
> > Q2. "It seems that we are just learning a robust policy that is robust to perturbation in actions rather than improving the training stability. Is this fixing training dynamics or just learning smoother policies?"
>
> Thank you for your question contrasts two interpretations of our method's effect. We argue that these two are directly linked: we are fixing the training dynamics by enforcing a smoother, more stable optimization landscape. The smoother policy is the result of this more stable training process, not a separate goal.
>
> We agree that our current figures do not provide a direct comparison to prove this. Figure 1 compares against a different regularizer, and Table 4 shows final performance, not the training dynamic.
> To provide direct evidence of improved training stability, we add a new experiment to the paper. We train our full PQL (Ours) and the unregularized PQL-PDE (w/o PDE) baseline over 8 random seeds and plot the learning curves (Normalized Score vs. Training Steps) with their standard deviation bands in Figure 2 in Page 10.
>
>
> > Q3 "In line 073, you did not modify the name of your method."
>
> We sincerely apologize for this placeholder. Since we determine the model name in the last minute, we forget to correct the template text in the contribution point. It has been fixed in the revised version.
>
> > Q4. "In Table 4, why do you think that beta sampling is improving the performance significantly?"
>
> Thank you for your question. The significant performance improvement from Beta sampling is not an independent gain; it is a synergistic enabler that is essential for our PDE regularizer to work. Our reasoning is based on the ablation study in Table 4 and is discussed in Section 5.4.
>
> First, our PDE regularizer ($\mathcal{L}\_{PDE}$) fundamentally changes the optimization problem. It constrains the model to find a solution that is not only accurate (imitation loss) but also geometrically simple (low Jacobian norm). This new, constrained problem is harder to solve than simple imitation.
>
> The proof is in the 'PQL-Beta (w/o Beta)' variant in Table 4. This model uses our PDE regularizer but with standard, naive uniform time sampling. Its performance is poor; the model "struggles across several environments". On Hopper-Medium-Expert, it scores only 98.7, which is worse than the 'PQL-PDE' model that has no regularizer at all (102.4). This result is critical: it demonstrates that simply adding the PDE constraint can be detrimental. As we state, "the unfocused updates from uniform sampling appear insufficient... within this... constrained space" in line 459.
>
> This is where our Beta sampling strategy becomes essential. It is specifically designed to solve this more difficult, constrained optimization problem. By sampling from a $Beta(\alpha, \alpha)$ distribution, we focus the training on the most informative parts of the generative trajectory. This provides the focused, efficient learning signal the model needs to find a high-performing policy within the new, smoother landscape created by the PDE regularizer.
>
> In summary, Beta sampling solely does not boost the performance (a finding consistent with prior work like FQL). Instead, it is the necessary partner to our PDE regularizer. The significant performance jump seen in Table 4 is the result of this synergy: the regularizer creates a better-structured solution space, and the Beta sampling provides the focused optimization tool required to find the optimal solution within it.

---

> ### Author Response · Authors · 2025-11-18
> **Author's Rebuttal Part Four**
>
> > Q5. What is the overhead of computing the Jacobian?
>
> Thank you for your comment. We do not compute the full, expensive $D \times D$ Jacobian matrix. As detailed in Section 4.1, we use Hutchinson's trace estimator. This method only requires Jacobian-vector products (JVPs), which are computationally efficient and have a cost roughly equivalent to one forward pass of the network. Since we use $K=1$ or $K=2$ probe vectors, the entire overhead of our regularizer is just $K$ additional forward passes per actor update. For your reference, we have attached the wall-clock training time compared with the standard Flow-matching RL baselines in two representative environments. While we would like to mention that implementation details may affect the wall-clock-time (due to different packages used), we are using the default implementations provided by the authors and did not make any further optimizations.
>
> | Model | Pen-human | AntMaze |
> | :--- | :---: | :---: |
> | FQL | 34:57 | 4:15:43 |
> | Flow | 35:10 | 5:21:50 |
> | CNF | 36:53 | 4:58:49|
> | PQL-PDE | 35:02 | 4:24:33|
> | PQL | 35:25 | 4:32:55|
>
> This new analysis provides a clear, empirical answer. The overhead of our $\mathcal{L}\_{PDE}$ regularizer is isolated by comparing PQL-PDE (our ablation baseline) to our full PQL method. This shows the actual cost of our contribution is minimal. We believe this is a very modest and practical trade-off for the significant performance and stability gains our method provides.
>
> We thank reviewer for the insightful feedback and are happy to provide any further clarification/justification if required.

---

> ### Author Response · Authors · 2025-11-27
>
> Dear Reviewer ZP6n,
>
> As the discussion period concludes in a few days, we would like to gently follow up on our response posted earlier.
>
> We would greatly appreciate it if you could review our rebuttal at your earliest convenience. Your feedback is very valuable to us, and we are keen to use the remaining time to address any further questions or concerns you might have.
>
> Thank you for your time and efforts.
>
> The auhtors of paper 5408.

---

### Official Review · Reviewer_Mgme · 2025-10-31

**Soundness:** 4
**Presentation:** 3
**Contribution:** 3
**Rating:** 8
**Confidence:** 3

**Summary:**

To address the issue in previous Flow-matching methods, where point-to-point optimization often neglects the global properties and smoothness of the generative path, this work introduces a regularizer based on partial differential equations (PDEs) to constrain the learning process. Additionally, a Beta-distributed time sampling strategy is proposed to improve the optimization efficiency of this regularizer. Experimental results on multiple offline RL benchmarks, including D4RL, Adroit, and OGBench, demonstrate the effectiveness of the proposed approach. Furthermore, the authors provide theoretical justification for the method.

**Strengths:**

1. The manuscript exhibits a coherent and logical structure, employs precise and formal language, and presents a clearly defined motivation that is readily understandable.
2. The study includes a wide range of rigorous experiments, offering insightful analysis of the outcomes and thorough evaluation of the model’s structure and hyperparameters.
3. The paper is grounded on a solid theoretical foundation, and the provided theoretical analysis offers principled support for the effectiveness of the proposed model.

**Weaknesses:**

1. The experimental baselines do not include comparisons with diffusion-based methods such as Decision Diffuser, Diffuser, or Diffuser-Lite, which would provide a more comprehensive evaluation.
2. The manuscript does not discuss the limitations of the proposed approach, which is important for understanding its scope and potential drawbacks.
3. A placeholder remains in the Introduction: the third point summarizing the contributions still contains “[Your Method Name]” and should be properly updated.

**Questions:**

1. Could the authors provide the performance results of their method on the Kitchen environment?
2. The proposed approach introduces a relatively large number of hyperparameters. While the experiments analyzing hyperparameters in the paper are appreciated, could the authors provide practical guidelines or recommendations for selecting hyperparameters to facilitate rapid adaptation to new tasks?
3. Have the authors considered alternative strategies to reduce the model’s sensitivity to hyperparameter choices?

---

> ### Author Response · Authors · 2025-11-18
> **Author's Rebuttal**
>
> Thank you for your valuable comments and suggestions. Please find our point-by-point response below.
>
> >  W1. "The experimental baselines do not include comparisons with diffusion-based methods such as Decision Diffuser, Diffuser, or Diffuser-Lite, which would provide a more comprehensive evaluation."
>
> We thank the reviewer for this comment.  We would like to clarify that our experimental evaluation does include a comprehensive comparison against several categort state-of-the-art (SOTA) diffusion-based models. As detailed in our Baselines subsection (Section 5.1), we benchmark PQL against strong, recent diffusion methods, specifically IDQL, SRPO, and CAC.
> We deliberately selected these models as they represent the current state-of-the-art in diffusion-based offline RL. Methods such as Diffuser and Decision Diffuser, while foundational, are earlier, trajectory-based planning approaches. The baselines we included (IDQL, SRPO, CAC) are more recent, generative policy models that have been shown in their respective literature to outperform these earlier methods.
> Therefore, by comparing against the strongest and most recent diffusion-based baselines, we believe our evaluation is comprehensive and provides a fair assessment of PQL's performance relative to the current SOTA.
>
>
> > W2. The manuscript does not discuss the limitations of the proposed approach, which is important for understanding its scope and potential drawbacks.
>
> Thanks for your comment. We have added the following limitations in the conclusion part:
> While PQL demonstrates strong performance, it has two primary limitations. First, our PDE-regularizer introduces a modest computational overhead, as it requires $K$ additional Jacobian-vector products (JVPs) per training step. Second, our method adds two key hyperparameters, the regularization weight $\lambda$ and the sampling parameter $\alpha$. While our extensive analysis shows these parameters are robust and consistent across a wide range of tasks, they still require initial tuning。
>
> > W3. A placeholder remains in the Introduction: the third point summarizing the contributions still contains “[Your Method Name]” and should be properly updated.
>
> We sincerely apologize for this placeholder. Since we determine the model name in the last minute, we forget to correct the template text in the contribution point.
>
> > Q1. Could the authors provide the performance results of their method on the Kitchen environment?
>
> Thank you for your suggestion. Please find the new results in the table below. Given the limited timeframe, we were not able to re-run all baselines on the Kitchen benchmark. However, we have run our own method (PQL) and are reporting its performance here.
>
> For comparison, we have included the results for CAC, as the authors of that paper also reported performance on this environment
> | Kitchen Tasks | CAC | **PQL (Ours)** |
> | :--- | :---: | :---: |
> | kitchen-complete-v0 | 51.9 $\pm$ 6.0 | **52.1 $\pm$ 3.5** |
> | kitchen-partial-v0 | 38.2 $\pm$ 1.8 | **40.1 $\pm$ 1.1** |
> | kitchen-mixed-v0  | 45.8 $\pm$ 1.5 | **43.9 $\pm$ 0.7** |
>
> Given the time constraints, we were not able to perfectly fine-tune our method on these new tasks. However, we believe these initial results are indicative of PQL's performance and stability.
>
> > Q2. The proposed approach introduces a relatively large number of hyperparameters. While the experiments analyzing hyperparameters in the paper are appreciated, could the authors provide practical guidelines or recommendations for selecting hyperparameters to facilitate rapid adaptation to new tasks?
>
> Thanks for your comment, as they are a direct result of our experimental analysis and the core, synergistic motivation of our method.Our process for finding these hyperparameters was guided by the two-part nature of our solution. We first introduced the $\mathcal{L}_{PDE}$ regularizer and quickly identified that a small, non-zero weight $\lambda$ is sufficient to stable the training process. A large $\lambda$ (e.g., 1.0) over-smooths the policy and hurts performance, while $\lambda=0$ is the unstable baseline.However, as our ablation demonstrates, simply adding this regularizer with standard uniform sampling (the 'PQL-Beta' variant) is detrimental to performance. We hypothesized this is because the regularizer creates a non-uniform optimization challenge, which is poorly matched to a uniform sampler. This led to our key insight for the sampling strategy. We recognized that the standard $Unif(0, 1)$ distribution is simply a special case of the Beta distribution: $Unif(0, 1) \equiv Beta(1, 1)$. This provided a principled, simple family of distributions to explore. Since the standard Beta distribution is $Beta(\alpha, \beta)$, we felt this would introduce too many hyperparameters. To simplify, we investigated the symmetric $Beta(\alpha, \alpha)$ family, where $\alpha=1$ is the known (suboptimal) uniform baseline.
>
> Q3 -  No, but we beleive it is a valuable direction for future research.

---

> > ### Comment · Reviewer_Mgme · 2025-11-24
> >
> > I appreciate the authors’ detailed responses to my questions and the effort put into additional experiments. Most of my concerns have been addressed.

---

> > > ### Author Response · Authors · 2025-11-25
> > >
> > > We are glad to hear that most of the concerns have been addressed. We appreciate the reviewer’s insightful comments and positive feedback.

---

### Meta-Review · Area_Chair_Wo4s · 2026-01-06

**Summary:**

This paper proposes PDE-regularized Q-Learning (PQL), which introduces a Jacobian regularization derived from the continuity equation to enforce smooth and stable flows. To optimize this objective, the authors adopt a Hutchinson's trace estimator and Beta-distributed time step sampling scheme. Experiments across diverse offline RL benchmarks show that PQL can improve performance by shaping the entire generative process rather than only the final policy.

The reviewers' main concerns include: (1) the Jacobian regularization is not new, (2) the performance improvement of PQL is marginal compared to many other baselines, (3) computational overhead of Jacobian regularization, and (4) lack of ablation studies.

**Reviewer Concerns:**

Reviewer ZP6n's concerns are (1) the Jacobian regularization is not new, (2) the performance improvement of PQL is marginal. I agree with this assessment and I don't think it is fully addressed.

Reviewer YF2m's concern is lack of ablation studies on Jacobian regularization and Beta-distributed time step sampling scheme. I believe the rebuttal does not fully address this concern.

Reviewer Y89o's concerns are (1) computational overhead of Jacobian regularization, and (2) ablation of Jacobian regularization and its effectiveness. (1) was addressed but (2) is not.

**Reviewer Scores:**

Reviewer Mgme's score is 8, and they have acknowledged the author rebuttal

Reviewer ZP6n's score is 2, and I don't think they will increase the score after author rebuttal

Reviewer YF2m's score is 4, and I don't think they will increase the score after author rebuttal

Reviewer Y89o's score is 6, and I don't think they will further increase the score after author rebuttal

---

### Decision · Program_Chairs · 2026-01-26

Reject